# Complexes of tubulin oligomers and tau form a viscoelastic intervening network cross-bridging microtubules into bundles

Phillip A. Kohl[1,12], Chaeyeon Song[2,3,4,5,8,12], Bretton J. Fletcher[2,3,4,5,12], Rebecca L. Best [4,6,9], Christine Tchounwou[2,3,4,5], Ximena Garcia Arceo[5,10], Peter J. Chung [2,3,4,5,11], Herbert P. Miller[4], Leslie Wilson[4,6], Myung Chul Choi[7], Youli Li [1] ✉, Stuart C. Feinstein[4,6] & Cyrus R. Safinya [2,3,4,5] ✉

The axon-initial-segment (AIS) of mature neurons contains microtubule (MT) fascicles (linear bundles) implicated as retrograde diffusion barriers in the retention of MT-associated protein (MAP) tau inside axons. Tau dysfunction and leakage outside of the axon is associated with neurodegeneration. We report on the structure of steady-state MT bundles in varying concentrations of $Mg^{2+}$ or $Ca^{2+}$ divalent cations in mixtures containing αβ-tubulin, full-length tau, and GTP at 37 °C in a physiological buffer. A concentration-time kinetic phase diagram generated by synchrotron SAXS reveals a wide-spacing MT bundle phase ($B_{ws}$), a transient intermediate MT bundle phase ($B_{int}$), and a tubulin ring phase. SAXS with TEM of plastic-embedded samples provides evidence of a viscoelastic intervening network (IN) of complexes of tubulin oligomers and tau stabilizing MT bundles. In this model, αβ-tubulin oligomers in the IN are crosslinked by tau's MT binding repeats, which also link αβ-tubulin oligomers to αβ-tubulin within the MT lattice. The model challenges whether the cross-bridging of MTs is attributed entirely to MAPs. Tubulin-tau complexes in the IN or bound to isolated MTs are potential sites for enzymatic modification of tau, promoting nucleation and growth of tau fibrils in tauopathies.

Microtubules (MTs) are hollow protein nanotubes resulting from GTP-mediated assembly of αβ-tubulin heterodimers, which stack to form curved GTP-tubulin oligomers and protofilaments (PFs). PFs interact laterally to form the MT wall[1], with these interactions hypothesized to stabilize their straight conformation[2,3]. Once incorporated in an MT wall, GTP at the β-tubulin subunit can hydrolyze, forming GDP-tubulin PFs that tend to adopt a higher curvature conformation often leading to MT disassembly[4]. These distinct PF conformations underlie MT

[1]Materials Research Laboratory, University of California, Santa Barbara, Santa Barbara, CA 93106, USA. [2]Materials Department, University of California, Santa Barbara, Santa Barbara, CA 93106, USA. [3]Biomolecular Science and Engineering, University of California, Santa Barbara, Santa Barbara, CA 93106, USA. [4]Department of Molecular, Cellular, and Developmental Biology, University of California, Santa Barbara, Santa Barbara, CA, USA. [5]Department of Physics, University of California, Santa Barbara, Santa Barbara, CA 93106, USA. [6]Neuroscience Research Institute, University of California, Santa Barbara, Santa Barbara, CA 93106, USA. [7]Department of Bio and Brain Engineering, Korea Advanced Institute of Science and Technology, 291 Daehak-ro, Daejeon 34141, Korea. [8]Present address: Amorepacific R&I Center, Yongin 17074, Republic of Korea. [9]Present address: Serimmune Inc., 150 Castilian Dr., Goleta, CA 93117, USA. [10]Present address: Department of Chemistry and Biochemistry, University of California, San Diego, San Diego, CA 93106, USA. [11]Present address: Department of Physics and Astronomy, University of Southern California, Los Angeles, CA 90089, USA. [12]These authors contributed equally: Phillip A. Kohl, Chaeyeon Song, Bretton J. Fletcher. ✉e-mail: youli@mrl.ucsb.edu; cyrussafinya@ucsb.edu

dynamic instability (DI) and enable the stochastic switching between periods of slow growth (polymerization) and rapid depolymerization[2–6], regulated in part by the relative abundance of GTP- and GDP-tubulin. In cells, MT-associated proteins (MAPs) can also regulate DI and are implicated in the formation of MT bundles[7–11], which are involved in numerous cellular functions[8,12,13]. Linear MT bundles (fascicles, Supplementary Fig. 1) with large wall-to-wall spacing are found in the central core of the axon initial segment (AIS) of mature neurons[14–16], which form a retrograde diffusion barrier for MAP tau, compartmentalizing tau in the axon[17].

In our study, we focused on mixtures of αβ-tubulin and the canonical, full-length isoform (4RL) of MAP tau, an intrinsically disordered protein that binds to MTs, partially suppresses DI, and facilitates the transport of cargo along MTs in axons[8,18–23]. Tau dysfunction is implicated in neurodegenerative tauopathies, which include Alzheimer's disease[24], FTDP-17[25], and chronic traumatic encephalopathy[26]. Humans express six wild-type tau isoforms resulting from alternative splicing of exons 2, 3, and 10 of the MAPT gene (Fig. 1A)[27]. The N-terminal projection domain (PD) and the C-terminal tail of tau protrude radially outward when bound to MTs, while tau's proline-rich region and MT binding region (MTBR) are enriched with cationic residues thought to interact with negatively charged residues at the carboxyl-terminal end of αβ-tubulin[28–30].

In-vitro experiments of paclitaxel-stabilized MTs in the absence of tau show that divalent cations induce ion-specific MT bundling[31] and depolymerization[32]. In follow-up studies excluding paclitaxel, tau-stabilized MTs reproduced linear bundles mimicking MT fascicles found in the AIS[33]. Building on those results, the current study seeks to elucidate ion-specific effects on the structure and stability of tau-stabilized MT bundles by adding $Mg^{2+}$ or $Ca^{2+}$ to minimal dissipative reaction mixtures at 37 °C containing αβ-tubulin, 4RL-tau, and GTP in standard PIPES buffer at pH 6.8[34]. The mixtures contained a 1/20 tau/tubulin-dimer molar ratio, which corresponds to sub-monolayer coverage of tau on MTs in the mushroom regime[35]. Divalent cation concentrations were in the millimolar range, approximating average cellular $Mg^{2+}$ content[36]. Thus, a motivation for the study was to reveal ion-specific effects between metal ions at physiological $Mg^{2+}$ concentrations in a minimal, cell-free model of MT fascicles of the AIS.

Time-dependent synchrotron small-angle X-ray scattering (SAXS) was used to monitor samples over 33 h (in the presence of excess GTP), revealing three distinct tubulin structural states. Kinetic phase diagrams generated from this data show that the tau-tubulin structural

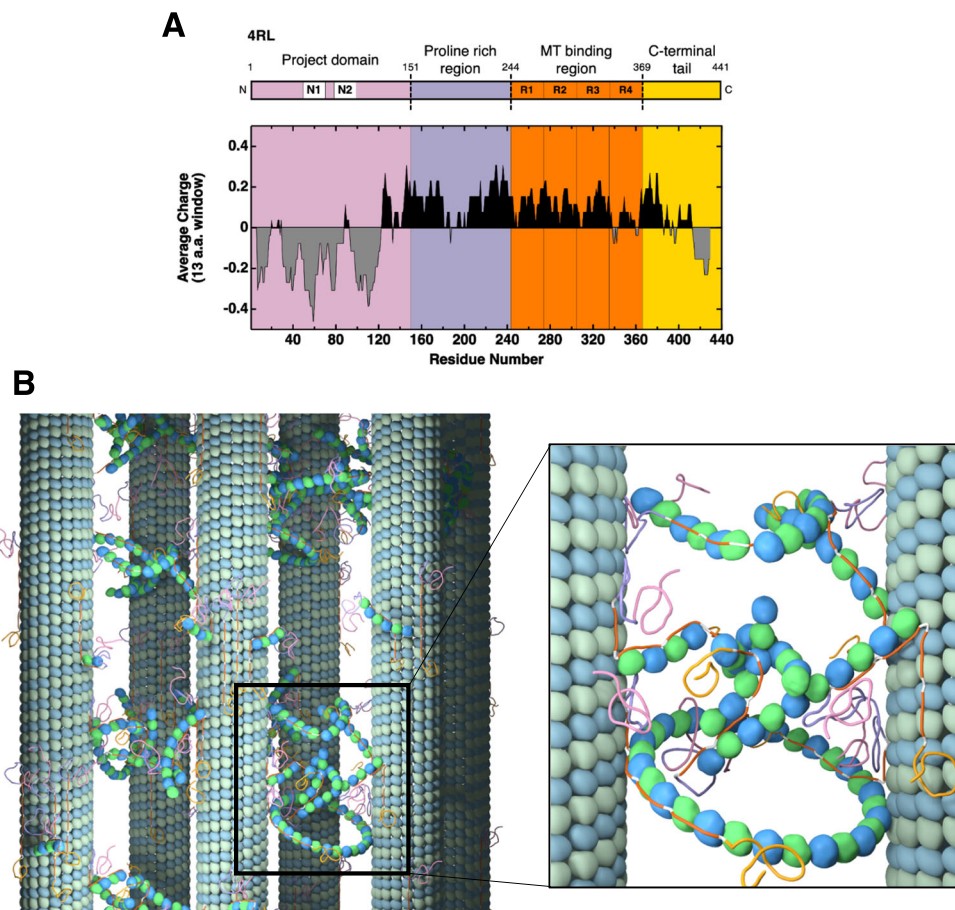

**Fig. 1 | Tau charge distribution and tau-tubulin network stabilizing bundled MTs. A** Schematic and average charge profile of full length 4RL tau with major features labeled, including inserts 1 and 2, encoded by exons 2 and 3, respectively, and all four MT-binding repeats (R1-R4). 4RL tau is depicted with labeled domains: N-terminal tail consisting of the projection domain (PD) and proline rich region, the microtubule (MT) binding region (MTBR), and the C-terminal tail[28–30]. The charge distribution is calculated using a rolling sum over thirteen residues. Despite anionic regions at the N- and C-termini, tau isoforms have a net cationic charge, which contributes to tau's binding (via MTBR) to negatively charged residues of αβ-tubulin. Data are from the National Center for Biotechnology Information Protein Database (accession number NP_005901.2). **B** Cartoon of a microtubule bundle (left) with blow-up (right) showing an intervening network of complexes of tubulin oligomers (curved short protofilaments and rings) and tau, which stabilizes MT bundles in the bundled wide-spacing ($B_{ws}$) and bundled intermediate ($B_{int}$) phases. The cationic MT binding repeats of tau (orange sections) link αβ-tubulin oligomers both to other free tubulin oligomers and to αβ-tubulin dimers within the MT lattice, creating the intervening network that cross-bridges neighboring MTs. Tau is depicted to bind either side of curved tubulin oligomers and rings consistent with studies that show tau may bind (via the MT binding repeats) either the outside surface or lumen of the MT[38,69,70].

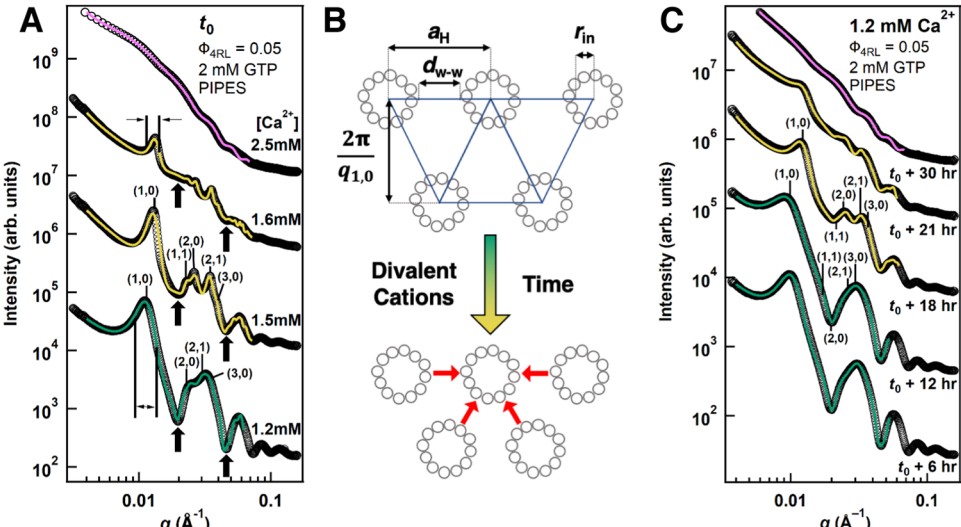

**Fig. 2 | Time-dependent synchrotron SAXS data with increasing divalent cations in PIPES buffer at pH 6.8 reveals an intermediate bundled ($B_{int}$) microtubule state between the bundled wide-spacing ($B_{ws}$) and the tubulin ring states. A** SAXS data (open circles) and corresponding fits (solid lines) for increased $Ca^{2+}$ concentrations at $t_0$, where $t_0$ is the time of the first measurement. Color of the fitted line represents the identified tubulin phase with green, yellow, and magenta, corresponding to profiles in the wide-spacing bundled ($B_{ws}$), intermediate bundled ($B_{int}$), and depolymerized tubulin ring states respectively. Indexing of two lower profiles is for a 2D hexagonal lattice of MTs. The $B_{int}$ (profiles at 1.5 mM and 1.6 mM $Ca^{2+}$) is differentiated from the $B_{ws}$ (profile at 1.2 mM $Ca^{2+}$) by peaks shifted to larger q (i.e. smaller lattice spacing), decreased peak widths

(compare 1,0 peak widths for 1.6 mM and 1.2 mM $Ca^{2+}$), and increased scattering intensity at local minima of the MT form factor (solid arrows at 1.2 mM, 1.5 mM, and 1.6 mM $Ca^{2+}$). At 2.5 mM of added $Ca^{2+}$, the SAXS is dominated by tubulin rings and curved oligomers. **B** Cartoon of hexagonal MT bundles in the $B_{ws}$ (top) and $B_{int}$, (bottom) states highlights changes in MT-MT spacing. **C** Time-dependent SAXS data (open circles) and corresponding fits (solid lines) of a sample with 1.2 mM added $Ca^{2+}$. Evolution of SAXS profiles show that phase transitions occur from $B_{ws}$ to $B_{int}$ (between $t_0 + 12$ and $t_0 + 18$ hrs) and from $B_{int}$ to the tubulin ring state (between $t_0 + 18$ and $t_0 + 30$ hrs). Source data for **A** and **C** are provided in the Source Data file.

state is a function of both time and $Mg^{2+}$ or $Ca^{2+}$ concentration (added to the PIPES buffer, Methods). Phase diagrams guided the selection of specific time points for parallel transmission electron microscopy (TEM) of plastic-embedded samples, providing real-space images for comparison to reciprocal-space SAXS data. In agreement with previous measurements by Chung et al.[33], widely-spaced MT bundles (labeled $B_{ws}$ with MT wall-to-wall spacing $d_{w-w} \approx 40$ to 45 nm), were stable below critical lower divalent cation concentrations $c_{lower}$ ($\approx 0.8$ mM $CaCl_2$ and $\approx 1.6$ mM $MgCl_2$ at 2 mM GTP). MT bundled structures also remained stable against added monovalent cations (up to 125 mM added KCl, above which SAXS data suggests KCl dramatically lowers tau tubulin binding affinity).

Here, we show that over a narrow range of added $Ca^{2+}$ or $Mg^{2+}$ concentrations ($c_{lower} < c < c_{upper}$ $c_{upper} \approx 1.4$ mM $CaCl_2$ and $\approx 2.4$ mM $MgCl_2$ at 2 mM GTP), the $B_{ws}$ state undergoes a complex structural evolution over a period of hours. The onset of this structural evolution is signaled by the depolymerization of a fraction of MTs, which leads to a sudden increase in the formation of tubulin rings and curved tubulin oligomers (observed in SAXS). During this period, remaining bundled MTs enter a transient intermediate bundle state ($B_{int}$), with a more ordered lattice and smaller $d_{w-w}$ ($\approx 30$–$35$ nm). In agreement with these SAXS findings, TEM images of MTs fixed in the $B_{int}$ state show larger, more tightly packed bundles, with smaller $d_{w-w}$, and significantly more cross-bridges between MTs (observed in both states as $\approx 5$ nm wide flexible filaments and rings), suggesting that the increased tubulin oligomer products created from depolymerized MTs distributed within the bundles, directly participate in the structural rearrangement of MT bundles by increasing the number of MT-MT cross-links. Taken together, SAXS and TEM data are consistent with MTs in both the $B_{ws}$ and $B_{int}$ states bundled by an intervening network of tubulin oligomers complexed with tau. This suggests a significant revision to current dogma where cross-bridges between MTs were attributed solely to MAPs[7–12,37]. In our model, MT bundle formation is due to coded assembly by tau's MT binding

repeats (orange sections in Figs. 1A and 1B) acting as the glue linking αβ-tubulin oligomers in the intervening network. In this role, tau links αβ-tubulin oligomers within the intervening network to one another and to αβ-tubulin within the MT lattice (Fig. 1B). Large MT wall-to-wall spacings observed in both bundled states (much larger than the size of tau's PD) are set by the average size of tau-coated, curved tubulin oligomers and rings.

## Results

### Time-dependent SAXS reveal three distinct tau-tubulin phases

To understand how divalent cation content modulates the stability and structural features of tau-mediated MT bundles, time-dependent synchrotron SAXS measurements (Supplementary Table 1) were performed on tubulin reaction mixtures (40 μM) containing 2 mM GTP, co-assembled at 37 °C with 4RL-tau (tau to tubulin-dimer molar ratio, $\Phi_{4RL} = 0.05$) at varying $CaCl_2$ or $MgCl_2$ concentrations (0 to 5 mM added to standard PIPES buffer containing 1 mM of $Mg^{2+}$ (Methods)). Reactions mixtures underwent 30 min of polymerization at 37 °C, followed by 30 min of centrifugation at 37 °C, and were then loaded onto the x-ray diffractometer with initial data points taken approximately 15 min after centrifugation (referred to here as timepoint $t_0$). Samples were exposed to 1 s of synchrotron radiation once every 3 h for a 33-hour period. Analysis of all azimuthally averaged SAXS profiles reveals three distinct concentration-dependent tau-tubulin structural phases outlined in Fig. 2A.

At initial timepoint $t_0$, all samples below a threshold divalent concentration of 1.4 mM added $Ca^{2+}$ or 2.4 mM added $Mg^{2+}$ exhibited MT bundling characteristics indistinguishable from controls with no added divalent cations (Supplementary Fig. 2). A typical SAXS profile representing the $B_{ws}$ phase is plotted in Fig. 2A and displays scattering characteristics indicative of strong MT polymerization and registers Bragg peaks consistent with 2D hexagonal packing of MTs ($q_{10}$, $q_{11} = 3^{1/2}q_{10}$, $q_{20} = 2q_{10}$, $q_{21} = 7^{1/2}q_{10}$, $q_{30} = 3q_{10}$, $q_{22} = 12^{1/2}q_{10}$). Wall-to-wall distances ($d_{w-w}$, Fig. 2B) calculated from the location of $q_{10}$ (Methods)

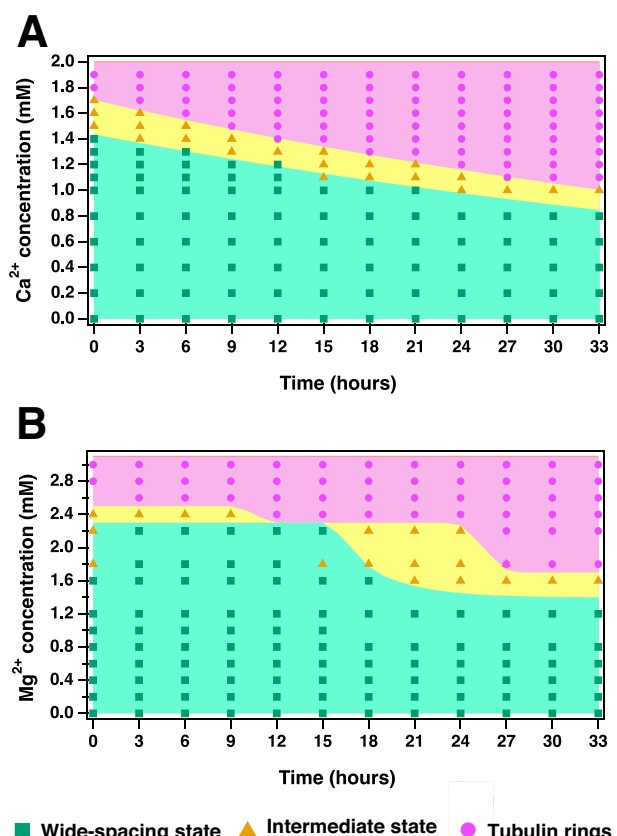

**Fig. 3 | Kinetic phase diagrams of tau/tubulin mixtures as a function of Ca²⁺ or Mg²⁺ concentration and time.** Markers denote the identified phase from line shape fitting analysis of SAXS experiments with green squares, yellow triangles, and pink circles corresponding to the $B_{ws}$, $B_{int}$, and (tau-coated) tubulin ring states respectively. Data shows distinct regions for each phase and highlights that the bundled microtubule state is a function of time and **A** Ca²⁺ or **B** Mg²⁺ concentration. Source data of all points is provided in the Source Data file.

ranged between $d_{w\text{-}w} = 29.5$ and $33.1$ nm for all samples, consistent with values of the widely-spaced MT bundle state reported previously[33].

Above threshold concentrations tau-tubulin reaction mixtures produced scattering profiles at initial timepoints that were markedly different from the $B_{ws}$ phase and are indicative of a new intermediate bundled state ($B_{int}$) (Fig. 2A, profiles at 1.5 mM and 1.6 mM Ca²⁺). One major difference between the two states is the characteristic shift in the location of all Bragg peaks to higher $q$ values, indicating that the wall-to-wall distance is smaller in the $B_{int}$ than in the $B_{ws}$ phase ($d^{int}_{w\text{-}w} = 24.5$ to $26.7$ nm). The Bragg peaks are also sharper in the $B_{int}$ state (Fig. 2A, full-width at half maximum (FWHM) for $q_{10}$, vertical lines at 1.2 mM and 1.6 mM Ca²⁺), implying that the coherent domain size of the MT lattice is much larger, despite having a smaller lattice parameter and decreased scattering from bundled MTs (Fig. 2A, scattering intensity of (1,0) peak diminishes with increased Ca²⁺). Unlike the $B_{ws}$ phase, scattering data shows that, within the $B_{int}$ phase, Ca²⁺ and Mg²⁺ inhibit MT polymerization and increase the prevalence of depolymerized tubulin products in a concentration-dependent manner, indicated by the decrease in bundled MT scattering and the increase in scattering at the form factor minima (Fig. 2A, arrows), respectively.

At higher Ca²⁺ and Mg²⁺ concentrations (1.9 - 3.0 mM Ca²⁺ or 2.6 - 3.0 mM Mg²⁺) MT polymerization was substantially inhibited, and scattering was dominated by single tubulin rings (Fig. 2A, broad oscillations at 2.5 mM Ca²⁺) and curved tubulin oligomers. Similar to previously reported free tau-tubulin heterodimer and oligomer

complexes[38–40] and tubulin spiral structures under non-assembly-promoting conditions[41], the SAXS data suggests that the curved tubulin structures in the three distinct states ($B_{ws}$, $B_{int}$, Ring) are coated with tau (discussed below).

Mg²⁺- or Ca²⁺-induced structural changes (smaller lattice size, larger bundle domain size, increased depolymerization) also occur as a function of time for samples with intermediate concentrations of added Ca²⁺ or Mg²⁺ ($c_{lower} \approx 0.8$ to $c_{upper} \approx 1.4$ mM Ca²⁺ or $c_{lower} \approx 1.2$ to $c_{upper} \approx 2.2$ mM Mg²⁺). As shown for 1.2 mM Ca²⁺ (Fig. 2C), each of these samples originates in the $B_{ws}$ state but abruptly transitions to the $B_{int}$ state after several hours. This transition is accompanied by increased scattering intensity at the form factor minima over time due to the rapid increase in MT depolymerization rate and tubulin ring proliferation, despite excess GTP remaining in the system for up to 72 h (Supplementary Fig. 2). Depolymerization of nearly all MTs and the structural evolution to the ring state is typically observed between 6 to 9 h after the $B_{ws}$ to $B_{int}$ transition. Figure 2C (1.2 mM Ca²⁺) shows a typical example, where the $B_{ws}$ phase is observed at $t_0 + 6$ hrs and $t_0 + 12$ hrs, the $B_{int}$ phase at $t_0 + 18$ hrs and $t_0 + 21$ hrs, and the ring state at $t_0 + 30$ hrs. We note that the observed depolymerization is not occurring due to denaturation from multiple one-second synchrotron exposures and that separate experiments testing the effects of prolonged X-ray radiation were unable to produce the intermediate state (Supplementary Fig. 3).

Kinetic phase diagrams for Ca²⁺ (Fig. 3A) and Mg²⁺ (Fig. 3B) summarize the SAXS data and visualize distinct regions where the $B_{ws}$ (green), $B_{int}$ (yellow), and (tau-coated) tubulin rings (magenta) are dominant. This data reveals a clear decrease in the lifetime of the $B_{ws}$ with increased divalent cation content, which is likely related to a similar effect by tetra-valent spermine on paclitaxel-stabilized MTs[42], where ion-induced depolymerization occurs due to disruption of the lateral bond between the M-loop on one β-tubulin and the H1-S2-loop on the neighboring β-tubulin. The intermediate state was not observed with increased monovalent cations (instead of divalent ions) added to the standard buffer, instead showing that $d_{w\text{-}w}$ remained constant over the tested range of added KCl (up to 150 mM KCl added to the PIPES buffer, Fig. 4).

We note that curved tubulin oligomers exist to some degree in all three labeled phases with a larger fraction in the $B_{int}$ state compared to the $B_{ws}$ state. Additionally, in the $B_{int}$ state, significantly increased scattering contributions from closed rings are observed. The ring state in Fig. 3 describes SAXS profiles where the $B_{MT}$ $q_{10}$ peaks are unable to be resolved. We also note that while the time of transition to the intermediate state was reproducible when comparing samples from the same batches of tau and tubulin, variability was observed when comparing the time of transition across different protein batches (Supplementary Fig. 4). Because of this, all time-based experiments presented here were conducted using the same batches of tau and tubulin.

Increasing (or decreasing) the initial GTP concentration for a prepared sample also effectively shifts the kinetic phase diagrams shown in Fig. 3 by either increasing (or decreasing) the time delay before the intermediate phase transition and by lowering (or increasing) the minimum divalent ion concentration necessary for the intermediate phase to be observed by 33 h. Figure 5 shows scattering profiles for samples prepared with 1.5 mM added Mg²⁺ (Fig. 5A) and Ca²⁺ (Fig. 5B) at time $t_0$ and varying GTP concentrations. Data shows clear examples of all three labeled tubulin phases with varying GTP concentrations and highlights that the concentration necessary to induce the intermediate phase is lower for Ca²⁺ compared to Mg²⁺.

While the time delay of the intermediate phase transition can be modulated with GTP concentration, it is important to note that the changes in the time of transition shown in Fig. 3 are primarily driven by the increased concentration in divalent cations and not due to differences in depletion or availability of GTP. This is apparent when

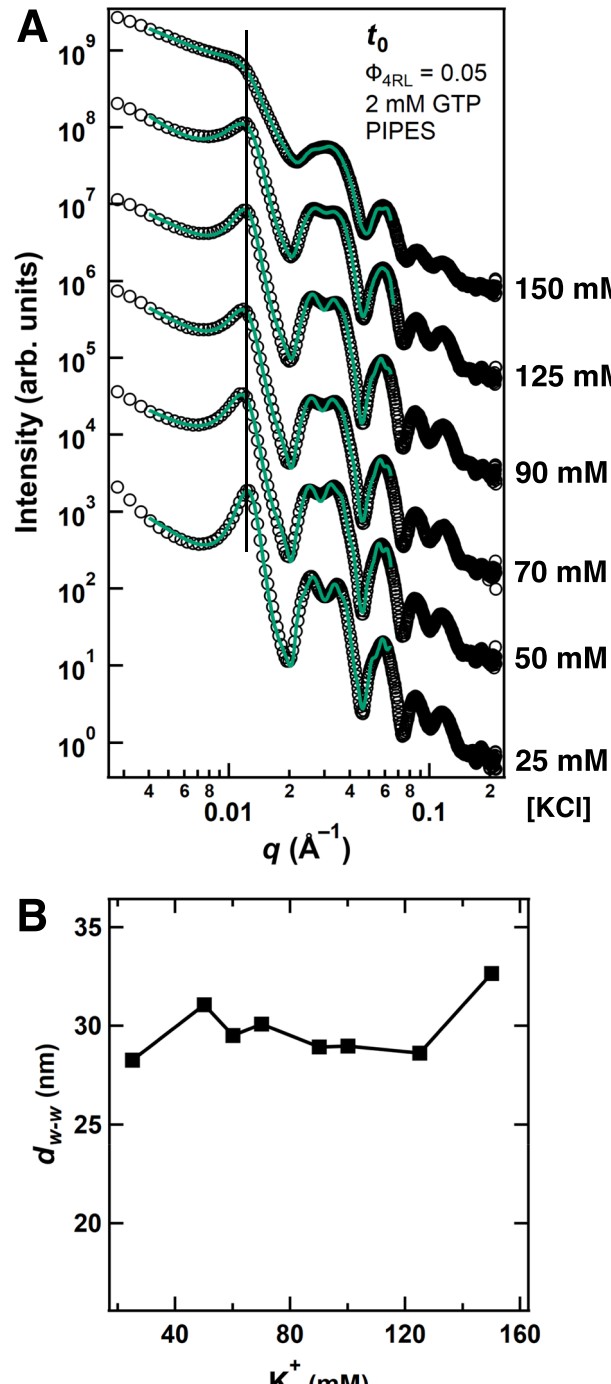

**Fig. 4 | Synchrotron SAXS data reveals the wall-to-wall distance of bundled microtubules is not dependent on KCl concentration. A** SAXS data (open circles) corresponding fits (solid green lines) of tubulin/tau/GTP mixtures at 37 °C and 4RL-tau to tubulin-dimer molar ratio $\Phi_{4RL} = 0.05$ with increasing KCl concentration. SAXS scans are offset for clarity. **B** Plot of the fitted wall-to-wall distance ($d_{w\text{-}w}$) from SAXS data in **A** highlights the lack of change in $d_{w\text{-}w}$ with increasing KCl concentration. Samples contained the stated KCl concentrations added to PIPES buffer at pH 6.8, which includes 1 mM of $Mg^{2+}$. All data is provided in the Source Data file.

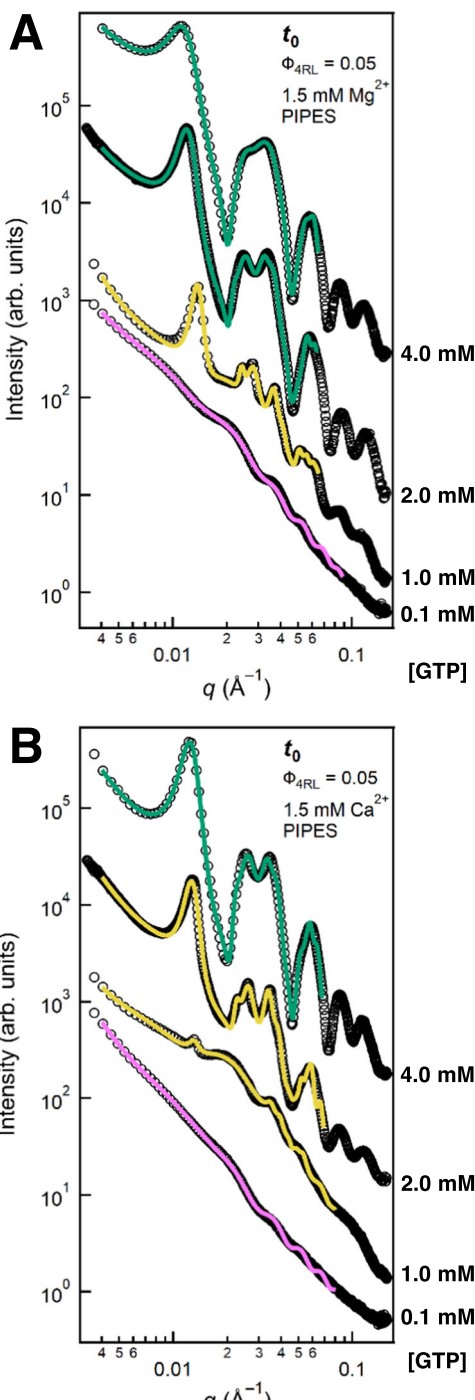

**Fig. 5 | Synchrotron SAXS data reveals that the tubulin-tau assembly state is GTP dependent.** SAXS data (open circles) and fit lines (colored solid lines) of tubulin/tau/GTP mixtures at 37 °C and 4RL-tau to tubulin-dimer molar ratio $\Phi_{4RL} = 0.05$ with increasing GTP concentration and 1.5 mM $MgCl_2$ (**A**) or $CaCl_2$ (**B**). SAXS scans are offset for clarity, and fit lines are color-coded to the dominant scattering phase for each sample with green, yellow, and magenta, corresponding to the wide-spacing bundled ($B_{ws}$), intermediate bundled ($B_{int}$), and depolymerized tubulin ring states respectively. Raw data and corresponding fits are provided in the Source Data file.

considering that GTPase activity at the β-tubulin subunit occurs primarily for GTP-tubulin incorporated in an MT lattice[43–46], limiting the rate of GTP hydrolysis proportional to the ratio of lattice-bound GTP-tubulin to free GTP-tubulin. SAXS data indicates this ratio is highest (most GTPase activity) at low divalent cation concentrations−where the scattering intensity from MTs is the highest. SAXS line-shape

analysis of samples with low divalent cation concentrations (below 1.2 mM $Mg^{2+}$ and 0.8 mM $Ca^{2+}$) indicates that MTs are stable at late time points (up to 72 h, Supplementary Fig. 1), despite high lattice-bound GTP-tubulin:free GTP-tubulin ratios, implying that GTP is present well beyond experimental timeframes, even when the rate of

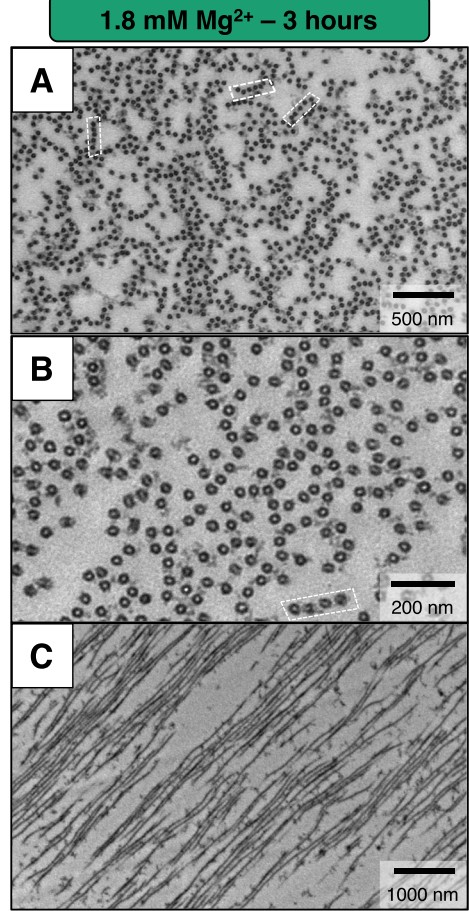

**Fig. 6 | Plastic-embedded TEM confirms the existence of the $B_{ws}$ and $B_{int}$ bundled phases.** Electron microscopy of microtubule assemblies prepared at 37 °C with mixtures of tau, tubulin, and 2 mM GTP in PIPES buffer at pH 6.8 with 1.8 mM added $Mg^{2+}$ and fixed after 3 h (**A–C**) and 18 h (**D–F**). Time points for sample fixation were selected based on SAXS data for samples prepared with identical conditions, where the wide-spacing $B_{ws}$ and the intermediate $B_{int}$ bundled phases were present at 3 and 18 h, respectively. **A, D** Top panels depicting cross-sections at low magnification show the extent of MT bundling for both phases ($B_{ws}$ state in **A** and $B_{int}$ state in **D**). The images further show the propensity for MT bundles to arrange in linear arrays (dashed white boxes). **B, E** Higher magnification cross-sections at

3-hour and 18-hour timepoints highlight the larger MT-MT spacing in the $B_{ws}$ compared to $B_{int}$ phase and the larger number of MT-MT cross-bridges per MT (and larger average number of neighbors for each MT) in the $B_{int}$ compared to the $B_{ws}$ phase. **C, F** Low magnification side views at 3-hour and 18-hour timepoints show that the width of the MT bundles (i.e. bundle size) is much larger in the $B_{int}$ compared to the $B_{ws}$ phase, and the spacing between MTs is smaller in the $B_{int}$ compared to the $B_{ws}$ phase, consistent with trends observed in SAXS data. All samples contained the stated $Mg^{2+}$ concentrations added to standard PIPES buffer at pH 6.8, which includes 1 mM of $Mg^{2+}$ (see Methods). Raw TEM images are provided in the Source data file.

hydrolysis is maximal. Thus, reaction mixtures with higher divalent cation content – where the $B_{int}$ state and the ring state are observed – must also contain excess GTP, even as MT depolymerization occurs.

## TEM confirms structural differences between $B_{int}$ and $B_{ws}$

To better understand structural differences between the two bundled states, TEM experiments were performed on plastic-embedded tau/tubulin/2 mM GTP reaction mixtures in standard buffer containing 1.8 mM of added $Mg^{2+}$ at 37 °C (Methods). Based on the phase diagram above, samples were individually prepared and fixed at 3 h (Fig. 6A–C) and 18 h (Fig. 6D–F) after polymerization to capture the $B_{ws}$ and $B_{int}$ states, respectively. Consistent with SAXS line-shape analysis, cross-sectional images at lower (Fig. 6A, D) and higher (Fig. 6B, E) magnification show distinct phase-separated bundled domains at both time points. Larger domain sizes (bundle widths) are seen at 18 h (Fig. 6D, E in $B_{int}$ state) compared to 3 h (Fig. 6A, B in $B_{ws}$ state), and measurements from higher magnification images also reveal an average inter-axial spacing ($a_H$) between bundled MTs that is 7.1 nm larger in the $B_{ws}$ phase (Fig. 6B) compared to the $B_{int}$ phase (Fig. 6E), comparable to values measured via SAXS ($\Delta a_H = 6.7$ nm, $\Delta a_H$ for TEM was determined

by comparing the Pair-Distance-Distribution-Function of $n = 4{,}843$ MTs and $n = 2{,}912$ MTs for $t_0 + 3$ h and $t_0 + 18$ h respectively). Similarly, TEM reveals MT pairs at 18 h have more cross-linkages and lower heterogeneity of wall-to-wall distances compared to those at 3 h, which coincides with narrower SAXS peak widths (i.e. larger coherent domain sizes) observed for the $B_{int}$ phase. Low-magnification side-view TEM images (Fig. 6C, F) also reveal structural features on length scales beyond the resolution of SAXS, showing that MT bundles in the intermediate state form larger extended arrays than in the wide-spacing state.

In agreement with plastic-embedded TEM measurements of Chung et al.[33] (with no added $Mg^{2+}$ to the buffer), few well-defined hexagonal arrays are observed, especially in the wide-spacing state. Instead, there is an apparent preference for MTs to form linear, string-like bundles (Fig. 6A, B, D, E white boxes) reminiscent of fascicles found within the axon-initial-segment (Supplementary Fig. 1). Even within regions of high microtubule density (Fig. 6D), where the probability of MT-MT interactions is higher, stacks of linear arrays or branched chains of MTs are more abundant than true hexagonal bundles.

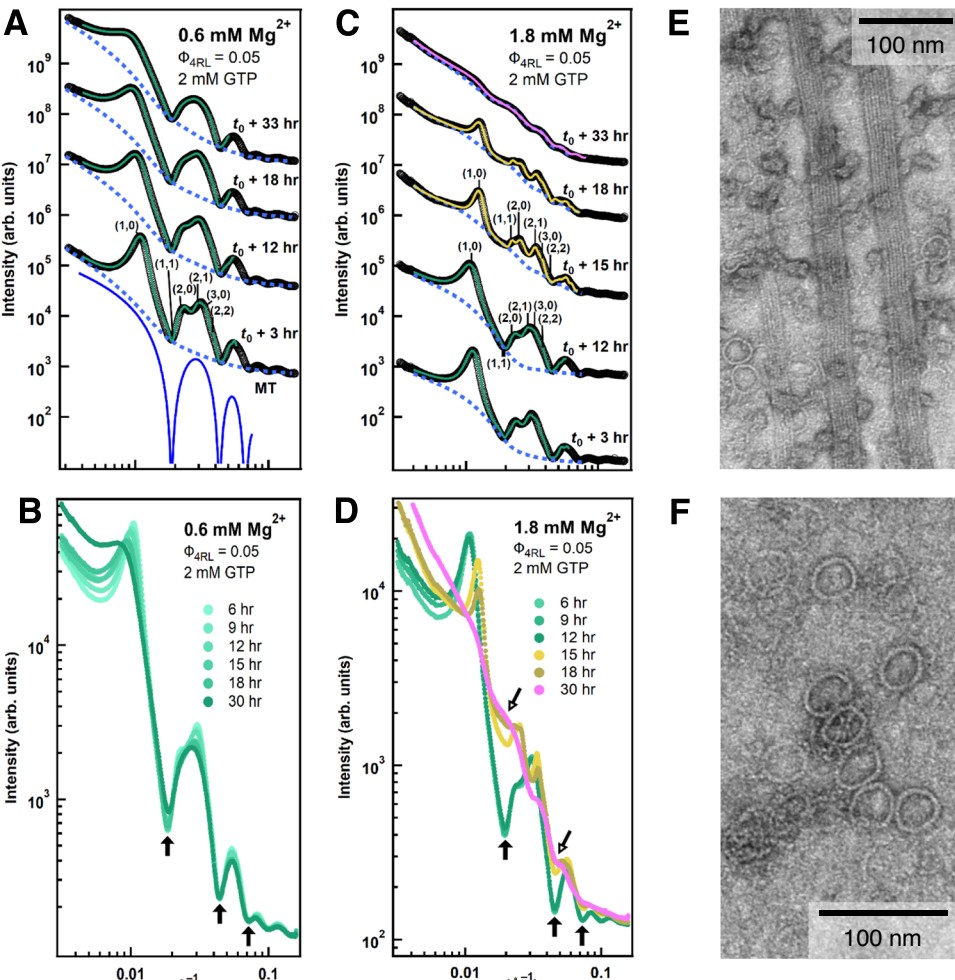

**Fig. 7 | Time-dependent synchrotron SAXS data at 0.6 and 1.8 mM Mg²⁺ in the wide-spacing (B$_{ws}$) and intermediate (B$_{int}$) microtubule (MT) bundle states reveal prevalence of rings in the B$_{int}$ state. A** SAXS data (open circles, profiles offset for clarity) and corresponding fits (solid lines) of a sample at 0.6 mM Mg²⁺ (below $c_{lower} \approx 1.6$ mM Mg²⁺) in the B$_{ws}$ state (2D Hexagonal peaks indexed for profile at $t_O + 3$ hrs). Dashed lines are non-MT scattering contribution obtained from fits. Bottom profile (solid blue curve) depicts the form factor of a MT. **B** Data from **A** plotted without offset. Solid arrows point to MT form factor minima, highlighting nominal change in non-MT scattering over time. (**C**) SAXS data (open circles, profiles offset for clarity) and corresponding fits (solid lines) at 1.8 mM Mg²⁺ (between $c_{lower} \approx 1.6$ mM Mg²⁺ and $c_{upper} \approx 2.4$ mM Mg²⁺) show a transition with increasing time from the B$_{ws}$ (profiles at $t_O + 3$ hrs and $t_O + 12$ hrs) to the B$_{int}$ (profiles at $t_O + 15$ hrs and $t_O + 18$ hrs) and to the tubulin ring state (profile at $t_O + 33$ hrs). 2D Hexagonal peaks indexed for profiles at $t_O + 12$ hrs (B$_{ws}$) and $t_O + 15$ hrs (B$_{int}$). **D** Data from **C** without offset. An abrupt increase in tubulin ring scattering fills in the minima of the MT Form Factor (open arrows) at the transition between $t_O + 12$ and $t_O + 15$ h. The fit lines in **A**–**D** are color-coded (green, yellow, and magenta represent the B$_{ws}$, B$_{int}$, and tubulin ring states, respectively). Parallel whole-mount TEM image taken at $t_O + 18$ hrs of a sample prepared with 1.8 mM Mg²⁺ added, showing **E** coexistence of tubulin rings with bundled MTs and **F** approximate dimensions of single tubulin rings. Samples contained the stated Mg²⁺ concentrations added to PIPES buffer at pH 6.8, which includes 1 mM of Mg²⁺. All data shown is provided in the Source Data file.

## Increased tubulin ring concentration in B$_{int}$ phase

To quantify the spike in MT depolymerization corresponding to the B$_{ws}$ to B$_{int}$ transition observed via SAXS (i.e. filling in of local minima, Fig. 2A, C), scattering data was fit to a model profile, $I(q)$ (Methods, Supplementary Code 1), which consists of three separate terms modeling hexagonally bundled microtubules[31,33,42], tubulin rings, and scattering from unpolymerized tau-tubulin mixtures and their aggregates plus a constant scattering background[47]. Separating the fit model into these three terms allowed us to differentiate the relative abundance of depolymerized tubulin in the ring configuration compared to unpolymerized tau-tubulin oligomers and aggregates (Methods). Comparing fit data from all of our samples revealed that the proportion of tubulin mass in the ring state is much higher in the B$_{int}$ compared to the B$_{ws}$ state.

Fits to the time-dependent SAXS data of reaction mixtures polymerized with 0.6 and 1.8 mM added Mg²⁺ (Fig. 7A–D) highlight the differences in scattering from unpolymerized tubulin oligomers between the B$_{ws}$ and B$_{int}$ states, similar to behavior found for Ca²⁺ samples (Fig. 2A, C). Solid lines through the SAXS profiles in Fig. 7A, C are fits of the data to $I(q)$ (Methods). For 0.6 mM Mg²⁺ (below $c_{lower} \approx 1.6$ mM Mg²⁺) the B$_{ws}$ state is stable for the duration of the experiment, despite indications of gradual MT depolymerization due to ongoing partially suppressed dynamic instability. Overlapping the SAXS profiles without offset shows a slight increase in scattering intensity at the first local minimum over time (Fig. 7B, solid arrows) but not at all other minima, indicating that tubulin rings were not formed (Methods) and that the tubulin depolymerization products are instead forming larger size tubulin oligomers and aggregates.

For Mg²⁺ concentrations between $c_{lower}$ and $c_{upper}$ ($\approx 1.6$ to $\approx 2.4$ mM Mg²⁺), the B$_{ws}$ state is stable for many hours with characteristically little tubulin ring formation (Fig. 7C, D, <15 h). However, transitioning to the intermediate bundled state coincides with a

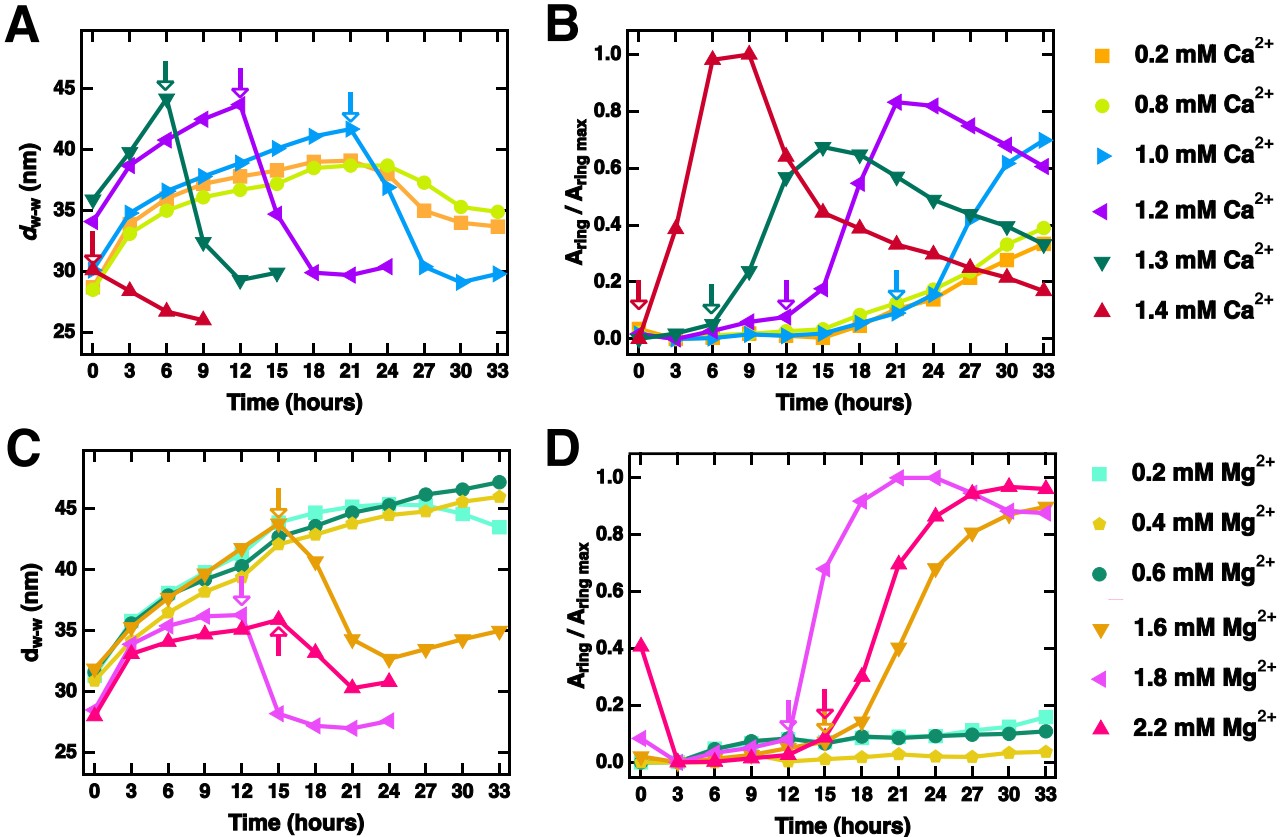

**Fig. 8 | Change in the microtubule (MT) wall-to-wall spacing, upon transitioning from the wide-spacing ($B_{ws}$) to the intermediate ($B_{int}$) MT bundle state, correlates with tubulin ring proliferation.** Wall-to-wall spacings ($d_{w-w} = a_h - 2[r_{in} + w]$) plotted as a function of time for the series of $Ca^{2+}$ (**A**) and $Mg^{2+}$ (**C**) samples (SAXS data shown in Figs. 2 and 7, respectively). Some data points omitted for clarity. Arrows indicate the latest time point at which the $B_{ws}$ state is observed before the sample transitions to the $B_{int}$ state. Fitted tubulin ring scattering amplitude plotted as a function of time for the same series of $Ca^{2+}$ (**B**) and $Mg^{2+}$ (**D**) samples. Amplitudes are normalized by the maximum ring scattering measured.

Arrows indicate the latest time point at which the bundled wide-spacing ($B_{ws}$) state is observed before the sample transitions to the bundled intermediate ($B_{int}$) state, with increases in the tubulin ring scattering amplitude observed after. Comparisons between **A** and **B** for the $Ca^{2+}$ series and between **C** and **D** for the $Mg^{2+}$ series show that the onset of tubulin ring proliferation occurs concurrently with the transition from the $B_{ws}$ to the $B_{int}$ state. Time "0" on the x-axis corresponds to $t_0$, as defined in Fig. 2. All samples contained the stated $Mg^{2+}$ or $Ca^{2+}$ concentrations added to standard PIPES buffer at pH 6.8, which includes 1 mM of $Mg^{2+}$ (see Methods). Data is provided in the Source Data file.

sudden spike in scattering intensity at all form factor minima (Fig. 7D, solid arrows). This distinct change in scattering (between 12 and 15 h for 1.8 mM $Mg^{2+}$) is well described by an increase in scattering from tubulin rings (Methods) and smaller curved tubulin oligomers. Plots of the raw scattering data for 1.8 mM $Mg^{2+}$ without offset (Fig. 7D) show a remarkable overlap of scattering intensity around the local minima at $t_0 + 18$ and $t_0 + 30$ hrs, with oscillations in SAXS at $t_0 + 30$ hrs well-described by the theoretical scattering profile of a single tubulin ring ($r_{in} = 17.3$ nm, Methods). Whole-mount TEM images prepared at 18 h (Fig. 7E, F) confirm this observation made with SAXS, showing the prevalence and coexistence of tubulin rings with MT bundles (Fig. 7E) which contain similar dimensions to those determined by SAXS (Fig. 7F). Together, our SAXS and TEM data directly show that changes to the bundled MT lattice strongly correlate with the sudden proliferation of tubulin rings and small curved tubulin oligomers (due to the dramatic increase in the rate of MT depolymerization).

SAXS data suggests that the coverage of tau on the MT surface remains roughly constant during MT depolymerization and thus implies that tau remains bound to the depolymerized tubulin products. This is inferred by measuring changes in the average MT's radius with time. Tau was previously shown to increase the average MT diameter in a concentration-dependent manner[48]. This effect was also observed through our own SAXS experiments designed to test the effect of tau coverage on MT wall-to-wall spacing (Supplementary Fig. 4), where tau-

tubulin dimer ratios ranging from $\Phi_{4RL} = 1/100$ (mushroom regime) to $\Phi_{4RL} = 1/5$ (brush regime) showed only small differences in $d_{w-w}$ between samples (Supplementary Fig. 4), but a monotonic increase in the MT radius. Conversely, throughout all time-based SAXS experiments where MT bundles were observed to transition from the wide-spacing to the intermediate state, the measured radius of the MT never increased but instead slightly decreased. Together this shows that the $d_{w-w}$ of the $B_{int}$ state cannot be explained by changes in the density of tau on the MT surface during MT depolymerization.

## $B_{ws}$ to $B_{int}$ transition coincides with tubulin ring formation

Figure 8 summarizes the results obtained from fitting SAXS data to $I(q)$. Within the $B_{ws}$ state (Fig. 8, arrows indicate the last time point where $B_{ws}$ was observed), $d_{w-w}$ increased rapidly at early time points due to relaxation from sample centrifugation and more slowly thereafter, reaching values up to $d_{w-w} \approx 49$ nm at 33 h (Fig. 8A, C). Unlike in the $B_{ws}$ state, $d_{w-w}$ spacings are comparatively stable in the $B_{int}$ state. The average stabilized wall-to-wall distance in the $B_{int}$ state for samples that transitioned during the experiment was 29.8 nm and was independent of $d_{w-w}$ before the time of transition. Similarly, samples initially observed in the intermediate $B_{int}$ state at $t_0$ (yellow regions at time = 0, Fig. 3A, B) show no relaxation from centrifugation over time (unlike the $B_{ws}$ state), instead converging to average spacing $d_{w-w} = 30.4$ nm for $Mg^{2+}$ and 26.4 nm for $Ca^{2+}$.

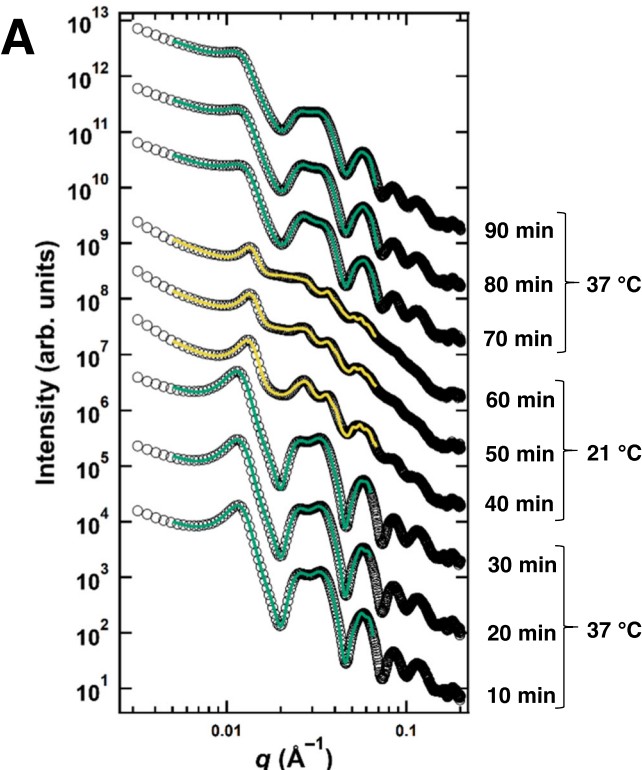

**Fig. 9 | Synchrotron SAXS data reveal the stability of the bundled wide-spacing state depends on temperature. A** SAXS data (open circles) and corresponding fits (solid lines) for a sample prepared with 2.0 mM $MgCl_2$ added to the standard PIPES buffer at pH 6.8. The sample temperature was held at 37 °C for 30 min, quickly reduced to 21 °C and held for 30 min, then quickly cycled back up to 37 °C. Data was taken via 1-second x-ray exposures every 10 min. Fit lines are color-coded for the wide-spacing bundled (green) and the intermediate state (yellow). Raw data and corresponding fits are provided in the Source Data file.

The stability of $d_{w\text{-}w}$ values measured in the $B_{int}$ state relative to those in the $B_{ws}$ state implies that stronger interactions dictate MT-spacing in the $B_{int}$ compared to the $B_{ws}$ state. This implication is further substantiated by the co-occurrence of the drop in $d_{w\text{-}w}$ and the substantial increase in coherent domain size (≈41 percent growth in the 6 h following the transition Supplementary Fig. 5) upon transitioning to the $B_{int}$ state (i.e. time between data collected at the arrows in Fig. 7A, C and the next data point).

For each sample, comparing $d_{w\text{-}w}$ to the amplitude of scattering from tubulin rings ($A_{ring}$, Fig. 8B, D) highlights the striking synchronization between the proliferation of tubulin rings and the abrupt drop in $d_{w\text{-}w}$. This reduced MT-MT spacing, together with the increase in MT bundle domain size, precisely when increasing amounts of rings and smaller curved tubulin oligomers begin proliferating, suggests that tubulin oligomers directly affect the bundling of MTs and drive the $B_{ws}$ to $B_{int}$ transition observed through SAXS and TEM.

### The phase of bundled MTs is tunable with temperature

To directly test the effect of enhancing unpolymerized tubulin oligomer content on MT bundling, time-dependent SAXS experiments were devised to modulate dynamic instability. Protofilaments (tubulin oligomers) in excess GTP are known to adopt a higher curvature conformation at low temperatures[49,50]. Therefore, lowering the temperature of the sample environment below a critical temperature for polymerization pushes the tubulin dynamic equilibrium toward depolymerization, effectively increasing the tubulin oligomer content in the solution.

A sample polymerized with 2.0 mM added $Mg^{2+}$ was monitored while the temperature was cycled for 30-minute periods at 37 °C, 21 °C, and 37 °C again. SAXS data from this temperature cycling experiment (Fig. 9) revealed that the otherwise stable wide-spacing state rapidly transitioned to the $B_{int}$ state immediately following the drop in temperature to 21 °C. The characteristic shift in hexagonal Bragg peaks to higher $q$ was observed, coinciding with a sudden increase in tubulin ring scattering. At 21 °C, $d_{w\text{-}w}$ decreased to a minimum value of 21.8 nm (change in wall-to-wall distance, $\Delta d_{w\text{-}w} = 8.5$ nm), and scattering from tubulin rings increased over the 30-minute incubation at 21 °C. Increasing the temperature back up to 37 °C following the 30-minute incubation at 21 °C, the microtubule bundles reverted to the wide-spacing state, as indicated by the increase in $d_{w\text{-}w}$ to 30.5 nm, decrease in the coherent bundle size, abrupt decrease in scattering from tubulin rings, and corresponding increase in $B_{MT}$ scattering. This result further demonstrates the correlation between free tubulin oligomer content and MT bundle architecture as the system transitions between the $B_{ws}$ and the $B_{int}$ states.

### TEM images show tubulin-tau complexes cross-bridging MTs

Plastic-embedded TEM side-views along the MT length provide potential insight into the structural components cross-bridging neighboring MTs and reveal a possible explanation for the $B_{int}$ phase observed through SAXS. Close inspection of bundles in the $B_{ws}$ phase at 3 h at lower (Fig. 10A, B) and higher (Fig. 10C–E) magnifications reveals an extended intervening network of crosslinked proteins connecting MTs (Fig. 10C–E). Remarkably, cross-bridges are seen not only connecting MTs within bundles (Fig. 10C–E, solid black arrows) but also connecting neighboring bundles (Fig. 10C–E, dashed black arrows). Evidence of these tethers is also seen within the high magnification cross-sectional views shown in Fig. 6, with more crosslinks present in the $B_{int}$ state (Fig. 6E) compared to the $B_{ws}$ state (Fig. 6B).

While cross-bridges between bundled MTs in cells have been previously reported but attributed only to tau[7,8,51], intrinsically disordered proteins such as tau (≈0.5 nm in width) are too thin to be visualized by TEM in our plastic-embedded preparations, implying that tau alone cannot make up the MT cross-bridge. In contrast, the morphology and dimension of the cross-bridges observed in TEM are consistent with tubulin oligomers, existing both as semi-flexible filaments (black solid and dashed arrows in Fig. 10C–E) and as tubulin ring structures (white arrows in Fig. 10C–E) with the same radius measured with SAXS and seen in whole-mount TEM (Fig. 7E, F). Thus, TEM data suggests that tubulin oligomers (which may include tubulin rings) are a central component of the observed intervening network between MTs and suggests that tau's role in MT bundling is to act as the glue (through the binding repeats) which connects tubulin oligomers.

## Discussion

The combined SAXS and TEM data lead us to propose a model where complexes of tubulin oligomers and tau form a viscoelastic intervening network that cross-bridges MTs into bundles. In this model bundling occurs due to coded assembly where tau's MT binding repeats link together αβ-tubulin oligomers in the intervening network and tubulin oligomers near the MT surface to tubulin in the MT lattice (Fig. 1B). The model presents an important revision to current dogma where cross-bridging between MTs is attributed entirely to tau[7,8,11,33,51–53].

This model is consistent with SAXS data for both the $B_{ws}$ and $B_{int}$ GTP-stabilized MT bundled states and the transition between them. The abrupt influx of (tau-coated) tubulin rings and curved tubulin oligomers (due to $Mg^{2+}$ or $Ca^{2+}$ mediated MT depolymerization of a fraction of MTs) drives the transition from $B_{ws}$ to $B_{int}$ by increasing the average number of tubulin-tau cross-bridges in remaining MT bundles (seen in TEM). Increased cross-bridging would explain the observed increase in bundle domain size of the $B_{int}$ state.

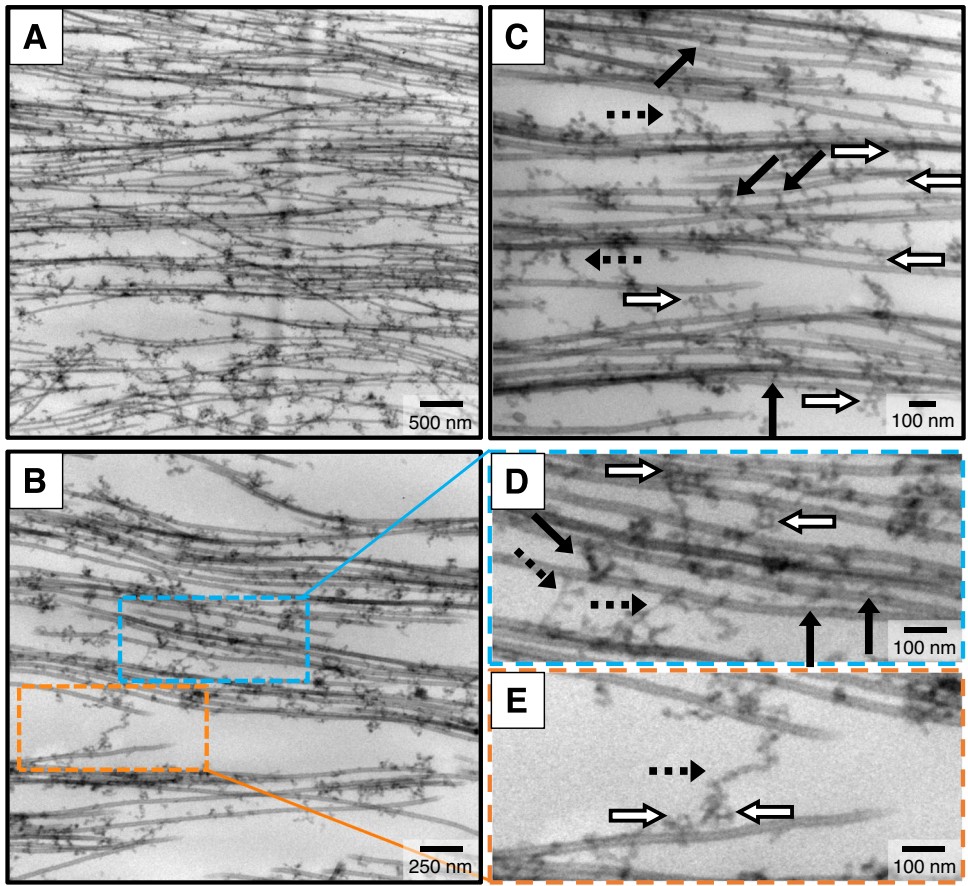

**Fig. 10 | Plastic-embedded TEM provides evidence that complexes of tubulin oligomers and tau cross-bridge bundled microtubules.** Electron microscopy with increasing magnification of microtubule assemblies prepared from mixtures of tau, tubulin, and 2 mM GTP in standard buffer with 1.8 mM added Mg²⁺ at 37 °C and fixed after 3 h. **A, B** At low and intermediate magnification, TEM images provide evidence that a network of filamentous proteins between bundled MTs is the linking medium that stabilizes MT bundles. **C–E** At higher magnifications the filamentous proteins of the network are seen to form MT-MT cross-bridges both within (black arrows) and between (black dashed arrows) bundled domains. The morphology of these cross-bridges is consistent with ≈ 5 nm wide semi-flexible tubulin oligomers. Tubulin ring structures (white arrows) are also present within the protein network. (**D, E** are expanded views of sections in B with blue and orange outlines). Complexes of tubulin oligomers and tau are also observed coating MTs but not forming MT-MT cross-bridges. All samples contained 1.8 mM Mg²⁺ added to standard PIPES buffer at pH 6.8, which includes 1 mM of Mg²⁺ (see Methods). Raw TEM images are provided in the Source data file.

The model reconciles years of contradicting reports regarding tau's role in bundling tau[7,8,11,51,54]. While numerous early publications pointed to MT bundling as one of many roles of tau[7,8,11,51], several cell-free studies of MTs containing the MT-stabilizing drug paclitaxel pointed to tau's apparent inability to mediate MT bundles[19,55]. More recent SAXS studies show that paclitaxel, at paclitaxel/tubulin-dimer molar ratios of $\Lambda_{paclitaxel} = 1/1$, suppresses MT bundling by all six tau isoforms[56]. Follow-up SAXS and TEM experiments showed that reducing paclitaxel below $\Lambda_{paclitaxel} \approx 1/8$ restores tau-mediated MT bundles[48]. These findings are consistent with our central discovery that bundling of MTs by tau requires free tubulin oligomers as $\Lambda_{paclitaxel} = 1/1$[56] severely reduces free tubulin oligomers and MT bundles, and reducing paclitaxel below $\Lambda_{paclitaxel} \approx 1/8$[48] restores free tubulin and MT bundles.

At high osmotic pressures, tau-coated MTs stabilized by paclitaxel form tightly packed bundles with $d_{w-w}$ spacings in the range of ≈ 3.5 - 4 nm—close to the radius of gyration of tau's PD[35] and consistent with polyelectrolyte theory[57]. In contrast, to account for the average large $d_{w-w}$ observed GTP stabilized MTs, tau-only models for MT bundling currently require a highly extended conformation for the projection domain (PD) of tau (Fig. 1A). Proposals of short-range charge-charge interactions between weakly penetrating tau PDs[33] and electrostatic zippers[53] based on the dipolar nature of tau's PD require extensions of the PD by factors of two and four, respectively,

(Methods). Such extensions of tau's PD are inconsistent with traditional polyelectrolyte theories[57] for chain stretching in the mushroom regime (at low tau/tubulin dimer molar ratio) and would require sequence-specific theories of highly stretched polypeptide chains at physiological salt concentrations (≈ 150 mM 1:1 salt).

In our model, the wall-to-wall spacing is largely set by the average radius of a mixture of (tau-coated) curved tubulin oligomers and tubulin rings (outer diameter ≈ 40 nm). However, the relative fraction of curved tubulin oligomers and rings are different in the $B_{ws}$ state versus the $B_{int}$ state, which contains significantly more rings. Quantitative SAXS line shape analysis indicates that the average size of free curved tubulin oligomers present in the $B_{ws}$ state is larger than in the $B_{int}$ state. This is consistent with the smaller wall-to-wall spacing of the $B_{int}$ state.

Because the binding interactions between tau and tubulin oligomers are specifically encoded by the MT binding repeats of tau, MT bundling is expected to be only weakly dependent on added monovalent salts. Indeed, $d_{w-w}$ is found to be essentially constant over a wide range of salt concentrations up to ≈ 150 mM KCl added to the buffer, which already contains ≈ 100 mM of 1:1 electrolytes (Fig. 4).

Together our results draw attention to the essential role of unpolymerized tubulin on MT bundling and highlight the significant impact of varied experimental conditions on tau's bundling function. Our finding that divalent cations near average physiological Mg²⁺

concentrations can destabilize dynamic, steady-state MT bundles at 37 °C, despite the presence of tau and excess GTP, is unexpected. In particular, at concentrations above a critical divalent concentration $c_{lower}$ (≈1.6 mM Mg$^{2+}$), MT bundles undergoing suppressed dynamic instability abruptly become unstable, favoring the formation of tau-coated tubulin rings and effectively halting dynamic instability. This suggests a mechanism where MT growth and stability can be modulated within cells through fluctuations in local divalent cation concentrations, with potential consequences for cargo transport in axons. Additionally, we expect the value of $c_{lower}$ to be modulated by disease-relevant post-translational modifications and truncations of tau.

We expect tau's role in MT bundling to be synergistic with other proteins that promote MT bundle formation in neurons, such as TRIM46[58–60], and other members of the vertebrate family of MAPs with similar MT binding regions[61]. The implications of our discovery of an intervening network of complexes of tubulin oligomer and tau should spur further studies on tau and other MAPs, such as MAP2, that mediate bundle formation with different wall-to-wall spacings[7,8,54,55].

A major function for bundled MT fascicles in the axon-initial-segment (AIS) is related to their role as a component of the filter for MT-based cargo trafficked between the soma and axon[17]. An intervening tubulin-tau network stabilizing MT fascicles should lead to a more efficient retrograde diffusion barrier that captures and prevents tau leakage outside of the axon. Biochemical alterations to tau (as happens in Alzheimer's disease and other tauopathies) could further modulate the barrier properties in the AIS, with significant implications for missorting of tau to the somatodendritic compartment and neurodegeneration. Finally, complexes of tubulin oligomers and tau of the intervening network, or those bound to MTs not in bundles, may represent an important site of chemical modification or fragmentation of tau by enzymes leading to aberrant tau behavior and nucleation and growth of tau fibrils in tauopathies.

## Methods

### Tubulin and Tau Purification

Tubulin was purified from bovine brain obtained at a commercial slaughterhouse from Mr. Ramero Carlos of Manning Beef LLC in Pico Rivera, CA. Purified tubulin was filtered into PEM 50 buffer containing 50 mM PIPES, 1 mM MgSO$_4$, 1 mM EGTA, and 0.1 mM GTP, pH 6.8 with ≈ 75 mM NaOH, and drop frozen into liquid nitrogen[34]. First, meninges, superficial blood vessels, and blood clots were removed at the slaughterhouse. Brains were then blended for 30 s at a low setting in a Waring Commercial Blender at a ratio of 1.5 mL of L-GNPEM buffer, pH 6.85, per gram of wet brain weight and subsequently homogenized in a motor-driven Teflon pestle/glass homogenizer operated at maximum speed (Tri R Stir R motor). Brain homogenate was then centrifuged at 32,500 x $g$ for 40 min at 4 °C using 50 mL centrifuge tubes, supernatant collected, and GTP was added to supernatant in dry form for 2.5 mM total concentration. This crude brain extract was subjected to one purification cycle of the following steps: glycerol-free polymerization (incubation at 30 °C for 30 min), centrifugation (at 45,000 x $g$ for 30 min at 30 °C, pellet containing MTs collected), depolymerization (dilution of collected pellet into cold L-GNPEM buffer and incubation for 40 min at 4 °C), and clarification (centrifugation at 45,000 x $g$ for 45 min at 4 °C, supernatant collected). 2.5 mM GTP was then added to the resulting tubulin-rich supernatant, and the supernatant was incubated at 30 °C for 30 min. The resulting suspension of microtubules was centrifuged at 45,000 x $g$ for 60 min at 30 °C, with the pellet collected and frozen in liquid nitrogen and stored at −70 °C. To remove any microtubule-associated proteins from the purified tubulin, frozen pellets were quick thawed at room temperature, then cooled on ice, resuspended in PEM50 buffer, pH 6.8, homogenized in a glass pestle, and incubated on ice for 20 min. The solubilized pellet solution was then centrifuged at 105,000 x $g$ for 1 h at 4 °C. The supernatant was collected, and 100 μM GTP was added to the solution, which was subsequently passed through a phosphocellulose column and eluted into 2–3 mL fractions. Fractions containing purified tubulin were pooled, placed on ice, and concentrated to between 7 and 14 mg/mL as measured by Bradford assay using BSA as a standard, then drop-frozen in liquid nitrogen to be stored as frozen beads at −70 °C.

cDNA expression vectors (pRK) encoding full-length (4RL) tau was gifted by Dr. Kenneth Kosik (University of California, Santa Barbara). Following standard procedures[62], Tau was expressed in BL21(DE3) pLacI cells (Invitrogen) with 18 h incubation in 250 mL of Luria broth (10 g of tryptone, 5 g of yeast extract, and 10 g of NaCl per liter of DI water) followed by 24 h incubation in 6 L of auto-induction media (10 g of tryptone, 5 g of yeast extract, 0.5 g of dextrose, 2 g of α-D-lactose and 5 mL of glycerol per liter of 25 mM NaHPO$_4$, 25 mM KH$_2$PO$_4$, 50 mM NH$_4$Cl, 5 mM Na$_2$SO$_4$ in DI water). Bacteria were harvested by centrifugation in a Sorvall RC-5B Plus centrifuge at 4,200 x $g$ for 10 min maintained between 4 °C and 10 °C. Bacteria resuspended in BRB80 buffer (80 mM PIPES, 1 mM EGTA and 1 mM MgSO$_4$) pH 6.8 with ≈120 mM NaOH, total 1:1 ion equivalent of 160 mM, Debye length, $\lambda_D$ ≈ 7.6 Å[63]) were lysed by passing through a French pressure cell three times at >900 PSI, subsequently boiled for 10 min, and then centrifuged at 20,200 x $g$ for 40 min. The supernatant was collected and passed over a phosphocellulose anionic exchange column and eluted with increasing concentration of (NH$_4$)$_2$SO$_4$ (up to 1 M) in BRB80. Tau-containing fractions were subsequently pooled and brought to 1.25 M (NH$_4$)$_2$SO$_4$, then further purified using hydrophobic interaction column chromatography (HisTrap Phenyl HP, GE Healthcare), eluted with decreasing concentration of (NH$_4$)$_2$SO$_4$ in BRB80. Fractions containing pure tau were pooled, then concentrated and buffer-exchanged into BRB80 by successive centrifugation cycles using Amicon Ultra-15 Centrifugal Units with MWCO = 10,000 (EMD Millipore, Darmstadt, Germany). Final tau stocks were stored at −80 °C until needed for experiments. Concentration was determined by SDS-PAGE comparison with a tau mass standard, the concentration of which had been established by protein mass spectrometry and stored at − 80 °C in BRB80 (80 mM PIPES, 1 mM EGTA, 1 mM MgCl$_2$, pH 6.8 with ≈ 120 mM NaOH, total 1:1 ion equivalent of 160 mM, Debye length, $\lambda_D$ ≈ 7.6 Å[64]).

### SAXS and TEM (Whole-mount and plastic-embedded) sample preparation

Reaction mixtures were prepared on ice in the following way. Purified tubulin (92 uM stock) was thawed, diluted into PEM50 (50 mM PIPES, 1 mM EGTA, 1 mM MgCl$_2$, pH 6.8 with ≈75 mM NaOH, $\lambda_D$ ≈9.6 Å), and mixed with solutions of GTP in PEM50 (100 mM stock), tau (37 uM stock in BRB80), and PEM50 buffer with additional MgCl$_2$ or CaCl$_2$ content (5–50 mM stocks). Final reaction mixtures contained 4.0 mg/mL tubulin in 50 uL of buffer ($\lambda_D$ ≈9.5 Å). Sample tubes containing reaction mixtures were placed in a 37 °C water bath for 30 min of polymerization to reach dynamic equilibrium. The MT-tau reaction mixtures were then prepared for experiments as follows.

For SAXS, reaction mixtures were directly loaded into 1.5-mm diameter quartz mark capillaries (Hilgenberg GmbH, Malsfeld, Germany) after polymerization. Capillaries were subsequently spun in a capillary rotor in a Universal 320R centrifuge (Hettich, Kirchlengern, Germany) at 9500 x $g$ and 37 °C for 30 min to form protein-dense pellets suitable for SAXS. Pelleted capillaries were then sealed and held at 37 °C in a custom-made, temperature-controlled sample holder for data acquisition.

Sample preparations for whole-mount and plastic-embedded TEM mirrored methods previously described[31,33,42] and were as follows. For whole-mount TEM, reaction mixtures were diluted into a warm buffer to 0.2 mg/mL tubulin and loaded onto highly stable Formvar carbon-coated copper grids (Ted Pella, Redding, CA), with excess solution wicked with Whatmann paper after 2 min. 1% uranyl acetate was added to the grid for 20 s and wicked off. Then, five drops of Milipore H$_2$O (18.2 MΩ) were added and wicked off. Grids were allowed to dry for 24 h at room temperature before imaging.

For plastic-embedded TEM, reaction mixtures were centrifuged to a pellet in microcentrifuge tubes at 9500 x $g$ at 37 °C for 30 min. The supernatant was removed, and pellets were fixed with 2% glutaraldehyde and 4% tannic acid overnight. Pellets were stained with 0.8% $OsO_4$ in PEM50 buffer for 1 h and subsequently rinsed four times with PEM50. Another stain of 1% uranyl acetate stain was applied for 1 h and rinsed with DI water. Fixed and stained pellets were subsequently dehydrated with 25/50/75/100% solutions of acetone in DI water for 15 min each. Pellets were then embedded in an epoxy-based low viscosity embedding media prepared by mixing 5 g of ERL 4221 (3,4 Epoxy Cyclohexyl Methyl 3,4 epoxy Cyclohexyl Carboxylate), 4 g of D.E.R. 736 (diglycidyl ether of propylene glycol), 13 g of NSA (nonenyl succinic anhydride), 0.2 g of EASE (PolyCut-Ease), and 0.2 g of DMAE (dimethylaminoethanol). Dehydrated sample pellets were infiltrated with embedding media, poured into flat embedding molds, held at 65 °C for 48 h to polymerize, and then cooled overnight before sectioning. Plastic-embedded samples were then cut to ~ 50-nm slices with a microtome (Ted Pella, Redding, CA) and transferred to Formvar carbon-coated copper EM grids.

## X-ray scattering and analysis

SAXS experiments were performed at beamline 4-2 of the Stanford Synchrotron Radiation Lightsource at 9 keV using a custom-made temperature-controlled sample holder. Experiments were performed at 37 °C unless otherwise noted. For the temperature ramp-down experiment, initial data was taken at 37 °C. Then the temperature controller was set to 5 °C, which was reached over <7 min. A needle temperature probe was inserted into a water-filled quartz capillary placed in the sample holder for instantaneous temperature recordings during the ramp down, approximating the apparent temperature for the samples. Additional information on data collection and experimental setup is provided in Supplementary Table 1.

Data from 2D scattering images was obtained with a Pilatus3 X 1 M 2D-detector and azimuthally averaged to create 1D scattering profiles. Data reduction was performed using Nika SAS[65], and quantitative lineshape analysis was performed using a custom-written function in C-plot (Supplementary Table 1). Scattering data was fit to the model profile, $I(q)$:

$$I(q) = \int\int S(\mathbf{q_r})|F_{MT}(\mathbf{q_r},\mathbf{q_z})|^2 + \int\int |F_{Ring}(\mathbf{q_r},\mathbf{q_z})|^2 + BG(q) \quad (1)$$

The first term in $I(q)$ consists of the structure factor $S(\mathbf{q_r})$ of the bundled MT lattice multiplied by the form factor of a MT ($|F_{MT}(\mathbf{q_z},\mathbf{q_r})|^2$) and is averaged over all orientations in q-space ($\mathbf{q_r}$, $\mathbf{q_z}$ are wavevectors perpendicular and parallel to the MT cylinder axis). The structure factor of the bundled MT state was modeled as the sum of square Lorentzians at every 2D reciprocal lattice vector $q_{hk} = q_{10}(h^2 + k^2 + hk)^{1/2}$ with amplitude $A_{hk}$ and peak width $\kappa_{hk}$:

$$S(q_r) = \sum_{h,k}\left[A_{hk}/\left(\kappa_{hk} + \left(\mathbf{q_r} - q_{10}\sqrt{h^2 + k^2 + hk}\right)^2\right)\right]^2 \quad (2)$$

The first three Bragg peaks ($q_{10}$, $q_{11} = 3^{1/2}q_{10}$, $q_{20} = 2q_{10}$) were individually fit, while all other peaks were thereafter fit simultaneously. To limit the number of fitting parameters, all simultaneously fit peaks were assumed to have the same peak width as $\kappa_{20}$ (the highest-order peak that was individually fit). The center-to-center distance between microtubules is given by $a_h = 4\pi/(q_{10}\sqrt{3})$, and the coherent domain size of the MT lattice (i.e. the hexagonal bundle width) is inversely related to the width of the structure factor peaks and is given by $L_{lattice} = 2(\pi\ln4)^{1/2}/\kappa_{10}$[42].

The form factors (F) of both MTs and tubulin rings were calculated by modeling them each as hollow cylinders with uniform electron density[35,56], wall width set to $w = 49$ Å[63], and lengths fixed at $L_{MT} = 20,000$ Å (larger than the resolution of our wavevector) and $L_{ring} = 42$ Å (consistent with electron microscopy data for single tubulin rings)[56]:

$$|F_{MT}|^2 \propto |\left[(\sin(q_z L_{MT}/2)/q_r q_z)\right]\left[(r_{in}+w)J_1(q_r(r_{in}+w)) - r_{in}J_1(q_r r_{in})\right]|^2 \quad (3)$$

$$\left|F_{Ring}\right|^2 = |A_{ring}\left[\left(\sin\left(q_z L_{Ring}/2\right)/q_r q_z\right)\right]\left[(r_{ring}+w)J_1(q_r(r_{in}+w)) - r_{ring}J_1(q_r r_{in})\right]|^2 \quad (4)$$

Here, $J_1$ is the Bessel function of order 1, $r_{in}$ and $r_{ring}$ is the ensemble-averaged inner radius of the MT and tubulin ring in Eq3. and Eq.4, respectively, and $A_{ring}$ is the scattering amplitude of the ring state. The MT cylinder's inner radius, $r_{in}$, was the only fit parameter in the MT form factor. The tubulin ring inner radius and length was fit for samples identified to be in the ring state. For samples in the $B_{int}$ or $B_{ws}$ states, the ring's inner radius and length was held at values fitted for that sample at later time points (i.e. if it transitioned into the tubulin ring state), or set to literature values ($r_{ring} = 16.3$ nm and $L_{ring} = 42$ A) if the sample never fully depolymerized.

Since the first order Bessel functions of the MT form factor ($J_1$ terms in Eq3.) equal 0 for distinct values of $\mathbf{q_r}$ (given $r_{in}$), the scattering contribution from MTs (both bundled and unbundled) to the total raw scattering profile will also approach zero at these points, resulting in the deep local minima observed in the theoretical scattering from a single MT (Fig. 6A, bottom blue curve). Therefore, since scattering from all tubulin within the MT lattice is suppressed at these distinct values of $\mathbf{q_r}$, the measured intensity at these points within the raw data must be almost entirely due to the scattering from unpolymerized tubulin oligomers plus the q-independent background ($|F_{Ring}|^2$ and BG($q$), Eq. 1). Because of this, an approximation for the concentration of tubulin rings can be made as: 1) any increase in the concentration of tubulin rings would require the intensity of all Bessel function minima to increase by a linear amount and 2) the exponential Porod scattering from larger tubulin structures decays much quicker than tubulin ring scattering, meaning that observed changes in the scattering intensity at higher q (>0.04) are dominated by changes in $A_{ring}$.

The third term in Eq.1, BG($q$), which accounts for scattering from unpolymerized tau-tubulin mixtures and their aggregates plus a constant scattering background, was modeled simply as a two-layer, unified fit function[47] at the Bessel function minima for samples at initial timepoints ($t_0$):

$$BG(q) = G\exp\left(\frac{-q^2 R_g^2}{3}\right) + B_1\left[\frac{\text{erf}(\frac{qR_g}{\sqrt{6}})^3}{q}\right]^{-P_1} + \exp\left(\frac{-q^2 R_g^2}{3}\right)B_2 q^{-P_2} + BG_0 \quad (5)$$

The first term, $G$, is Guiner's law, the second and third terms, $B_1$ and $B_2$, are power-law scattering terms which describe the scattering at lengths where $qR_g \gg 1$ and $qR_g \ll 1$ ($R_g$ defined below), respectively, and the fourth term, $BG_0$, is a flat scattering term independent of $q$. Scattering from the third and fourth term in the equation above are only appreciable to the total scattering profile ($I(q)$, Eq. 1) at very low-q (<0.005 Å$^{-1}$) and high-q (>0.15 Å$^{-1}$) respectively, and thus were individually fit within these two domains. Fitting bounds for $R_g$ were determined by fits of the unified scattering function to unpolymerized tau tubulin samples consisting of a distribution of tubulin oligomers and their aggregates resulting from the presence of tau (with minimal scattering contributions from tubulin rings) and were fit simultaneously with the first Bragg peak, as its scattering was most prominent at this length scale ($R_g = 15$–20 nm). Lastly, the fitting parameters of the second term dominate the BG scattering signal for our samples between $q = 0.03$–0.09 Å$^{-1}$, and were thus determined by fitting it to the set of points measured at time $t_0$ where the scattering contribution

of the bundled MT state to the raw data was approximately zero (i.e. at the Bessel function minima).

## Transmission Electron Microscopy

All data were taken at 80 kV using the turn key JEM 1230 (JEOL) Transmission Electron Microscope at the University of California, Santa Barbara.

## Calculation of $R_G$ and projection domain extensions

Previously, the radius of gyration ($R_G$) of wild-type tau in solution was found[66,67] to scale as an unstructured protein with random coil behavior, with $R_G = 0.1927N^{0.588}$ nm, which was subsequently used to calculate $R_G^{PD} \approx 3.7$ nm and the physical diameter $D_{Phys} = 2R_{Phys} = 2(5/3)^{1/2}R_g \approx 9.5$ nm[68] for the PD of 4RL tau. For the weakly penetrating tau-tau interaction model, $D_{Phys}$ is doubled to account for each opposing tau PD, giving a predicted $d_{w\text{-}w}$ of 19 nm, roughly half the $d_{w\text{-}w}$ we observed via SAXS. Similarly, for the electrostatic zipper model that proposes complete overlap of two opposing tau molecules, the predicted $d_{w\text{-}w}$ would just be $D_{Phys}$, requiring significant extensions beyond previously reported $R_G$ values.

## Statistics and reproducibility

As described in the results section, the time of transition from the $B_{ws}$ to $B_{int}$ phase was consistent for samples prepared from the same batch of tau and tubulin. The time of transition at a given concentration across different batches however could differ by multiple hours (Supplementary Fig. 4). The general trends outlined within this paper however were consistent with all batches used. All data presented with the exception to the temperature experiments were performed using the same batch of tau and tubulin.

The number of times each of the SAXS experiments were independently conducted (across different batches of tau and tubulin) are as follows: thrice for Figs. 2, 3, 7, and 8, twice for Fig. 4, and once for Figs. 5 and 9.

For the plastic-embedded TEM, images came from 3 unique plastic-embedded slices for each time point. From them, 136 unique images of the $B_{ws}$ state were taken down the cylindrical MT axis and 34 unique images along the length of the MT. For the $B_{int}$ state, 57 unique images were taken along the length of the MT and 209 unique images were taken down the cylindrical axis.

Whole-mount TEM images presented are from a single independent experiment, however the experiment was independently conducted on five different occasions with similar results. From the experiment presented 46 unique images were taken at 3 h and 57 unique images were taken at 18 h.

## Reporting summary

Further information on research design is available in the Nature Portfolio Reporting Summary linked to this article.

# Data availability

The uncropped TEM images, SAXS data, and all resulting fits generated in this study are provided in the Source Data excel file. Source data are provided with this paper.

# Code availability

The code used for the fitting of the SAXS data can be found in the Supplementary Code 1 file.

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

## Acknowledgements

This work was supported by the Department of Energy, Office of Basic Energy Sciences, Division of Materials Sciences and Engineering, under award DE-FG02-06ER46314 (CRS and YL, charged bio-assemblies inspired by nature) and, in part, by the US National Science Foundation, Division of Materials Research, under award DMR-1807327 (CRS, phase behavior in biomaterials). PJC acknowledges support from the US National Science Foundation under award DMR-2104854. SCF acknowledges support from the US National Institutes of Health grant NS-35010 and the Academic Senate of the University of California at Santa Barbara. MCC was supported by Korea Dementia Research Center KDRC HU23C0094, KAIST Grand Challenge KC30 N11230021, Korea Basic Science Institute KBSI 2021R1A6C103B422, and National Research Foundation of Korea NRF RS202300277142 and NRF 2020M3A7B6026565.

## Author contributions

C.R.S., C.S., B.J.F., P.A.K. and Y.L. designed research; H.P.M. purified tubulin; C.S., B.J.F. and C.T. provided tau protein isoforms from plasmid preparations; C.S., B.J.F., P.A.K. and X.G.A. performed research; P.A.K. analyzed SAXS data; P.A.K., C.R.S., B.J.F. and Y.L. modelled data; P.A.K., B.J.F. and C.R.S. wrote the paper; Y.L., P.J.C., R.L.B., S.C.F., M.C.C., L.W., C.T. and H.P.M. provided additional writing input; and Y.L., S.C.F., M.C.C. and L.W. provided critical suggestions on the presentation of X-ray and TEM microscopy data.

## Competing interests

The authors declare no competing interests.
