## [Peer Review File · Nature Communications]

Complexes of tubulin oligomers and tau form a viscoelastic intervening network cross-bridging microtubules into bundlesReviewers' Comments:

Reviewer #1 (Remarks to the Author):

Review of: "Complexes of tubulin oligomers and tau form a viscoelastic intervening network cross-bridging microtubules into bundles", authors: Kohl et al.

This manuscript reports on the structure of microtubule (MT) bundles in the presence of tau protein under varying ionic conditions, analyzed as in vitro reconstituted systems by small-angle X-ray scattering and electron microscopy. MT bundles are essential components of many cellular machineries, including the mitotic and meiotic spindles, cilia and flagella and prominently as the core of axons and dendrites of neurons. MT-binding proteins, in particular tau, are implicated in neurodegenerative diseases. A multitude of associated proteins modify and bind MTs and control their dynamic functions in ways that are rather incompletely understood. Tau protein with its projection domains was, in an earlier publication by the same group, shown to promote stable MT bundling with 30-40 nm spacing. Here the authors show that 2 different stable bundled phases occur when divalent cation concentrations are varied. Importantly this work proves that tubulin oligomers and rings are necessary to stabilize wide-spaced bundles in the intermediate phase. This solves a puzzle about how the relatively short projection domains of tau could stabilize such bundles. This is a very interesting and thorough study of a mesoscopic assembly of microtubules that appears to be physiologically relevant. I have only a few minor comments (see below), and do think that this work will be of high interest for the broad readership of Nature Communications. I therefore recommend acceptance after some additional clarifications.

In detail:

- One major question in my mind is if the findings that are presented here from in vitro experiments are representative of what is happening in cells. Is there any evidence for this kind of ring- or oligo-based crosslinking in the axons of neurons?
- For clarification: how does the bundling observed here relate to microtubule organization by neurofilaments in axons?
- Could the authors discuss a bit more how they think the rather disordered crosslinking by tau and tubulin oligos and rings leads to a preferred distance between the microtubules as observed in SAXS? I could imagine smaller spacing with short oligos or large spacing with a clump of oligos or rings in between.
- Microtubule depolymerization is a physiological part of dynamic instability, but can also occur through protease activity or other degradation. During tubulin preparation from tissue, the whole purification procedure uses selection of the polymerizable fraction from all the rest, but that selection is of course never perfect. How reproducible were the results using different batches of tubulin from different preparations?
- In the time dependent SAXS experiments, is there any danger of prolonged X-ray exposure causing damage in the samples or were the samples not exposed repeatedly?

-

Smaller issues:

- Fig 1B: the color coding of the tau filaments is hard to see.
- Axis labels in the figures are varying widely, some are really small.

Some minor spelling issues I happened to see:

- Line 51: should read "...interactions are hypothesized.."?
- Line 506: should read "..Bradford..."
- Line 612: should read "...Guinier's law..."

Reviewer #2 (Remarks to the Author):

The authors characterized MT bundling with in situ small angle x-ray scattering and found that it changes continuously with time. Based on this finding, they claim the necessity of operating within physiologically relevant experimental conditions. More importantly, they proposed the role of tau in the MT bundling.

Their findings, I am pretty sure, are interesting to the community and may bring new insights in understanding the MT bundling. But, in my opinion, the authors make a conclusion rather too early without a strong support from data.

Data clearly shows the Bws transitions to the Bint and rings upon addition of divalent ions, temperature decrease, and time progress. However, it is hard to for me to accept the conclusion that the Bint requires free tubulins. Figure 4 shows that the first minima goes up, based on which the authors claims the formation of rings, but the Bint doesn't form in this case. So, doesn't the increase of the minima in this case indicate the increase of free tubulins?

They claims that GTP concentration or depletion is not a factor, then what happens if they uses much smaller amount of GTPs? or why not present data with less GTPs?

The authors did not put all the parameters obtained from the fit. Obviously, data of rings shows much finer oscillations and shift of minima toward smaller q region, suggesting a larger ring size. And the background increases more at smaller angles, for example, at the first minima and lower q. Is this suggesting the formation of a larger object instead of free tubulins?

Figure 6 may indicate that the SAXS invariant may stay the same, independent on time. If bundles become free tubulins and there's no loss of tubulins, this looks right in the invariant point of view. But, data shows increase of larger structures instead of increase of scatterings from smaller object, for example, free tubulins dimers. How can this be explained?

In biomolecules/biopolymers, upon the addition of divalent ions, the sequence of structural transitions,

free particles/chains-crystalline assemblies-free particles/chain, is commonly observed. Isn't the authors observation consistent with those?

Why the ionic effect is much slower than the temperature effect? The free tubulins formation may be important factor, but there needs some explanation about this kinetic effect. According to Ca^{2+} data from Figure 3A, tubulin rings may form at even low Ca^{2+} concentration by just waiting for long enough.

Please, do not omit experimental details. For example, what is t_0 ? The authors define it as a start of measurement, which seems too vague. It should be something like the time of mixing divalent ions. What is the plastics used for embedding? What are used for the A_{hk} in equation 2 and A_{ring} in equation 4?

Reviewer #3 (Remarks to the Author):

The main claim of the paper is that the tau-induced bundling of MTs involves crosslinks between them that include tau engaged with tubulin oligomers/rings, and that this is relevant to the in vivo case. However, there is no solid, direct proof that this is the case, and the authors will need to do more experiments to demonstrate that their claims are true, even in vitro.

There is no doubt that the authors have collected an impressive number of data points, but whether they are measuring anything of relevance is not so clear. The one solid conclusion that I can drive from their experiments is that something happens under conditions of high Ca or Mg concentration that leads to MT depolymerization into tubulin rings and oligomers, which occurs faster the higher the concentration of those ions, while there is an increased "compaction" of MT bundles. The authors deduced causality from the time coincidence, saying that tubulin oligomers/rings cause that tightening of the bundles. There are ways in which the authors could more directly test this hypothesis. For example, if soluble GDP-bound tubulin were to be added, would that speed up the formation of the more compact bundles? Because of the nature of their experiments, MT need to depolymerize to get the excess rings that the authors claim causes the tightening of the MT arrays, but ultimately this leads to full depolymerization of the MTs. If all that is needed is extra tubulin in an unpolymerized state to cause the tightening of the bundles, then adding GDP-tubulin to the MT-tau mixture under conditions that promote wider spacing should lead to their compaction without disassembly. Their temperature experiment still relies on a depolymerization process that ends with the disappearance of the MTs. Additionally, temperature can have many effects beyond MT depolymerization, for example, on the dynamics/conformation of tau.

The authors state: "we expect these curved tubulin structures to be coated with tau" (line 185-6). Indeed, it is an "expectation" of the authors, without a solid demonstration. The interpretation of the SAXS data and EM in terms of cross-bridges of tubulin oligomers-bound tau needs corroboration that is more direct. As it exists, the SAXS is not sufficient to support this proposal by itself, even if it may be

compatible with it, and the EM shows densities that are more easily explained by aggregated protein than tubulin oligomers/rings. The model may have some appeal, but it is in no way proven by the data presented.

The authors state “TEM data provides direct evidence that tubulin oligomers...”, but that is not true. What the extra density is, multiple tau molecules, tubulin oligomers or tubulin/tau aggregates of denatured material, it is not possible to say. The EM images provided are, by their nature, very low resolution. In this day and age, the authors should find ways to pursue imaging using cryo-EM, or more specifically, cryo-electron tomography. Their new model is radical enough to require the test of serious proof.

The authors state that the MT spacing in bundles is much larger than the tau PD, but fully extended PD and PRD will be able to cover those distances. They provide information for experimental radius of gyration for tau molecules under different conditions that invokes a significant compaction of an extended chain to about 40 Å (this is approximately the size of a tubulin monomer, when just one pseudo-repeat of tau extends over two when bound to a MT). That length is not compatible with binding to tubulin oligomers/rings. I do not see how the authors can use those numbers to justify that the distances between MTs cannot be explained by tau by itself, but then assume that those numbers are compatible with binding to multiple tubulins in oligomers and to MTs simultaneously! Have they obtained data about the radius of gyration of tau in the presence of tubulin rings alone, say, generated by GDP tubulin at low temperatures? Are those more compatible with the MT-MT distances they observe?

Generally, the paper is very descriptive and lacks molecular mechanistic insights. Is the compaction affecting the number of repeats that are attaching to MTs per tau molecule? The amount of oligomers bound to tau? What is the molecular rate limiting step that makes the evolution with time so slow?

Another problem with this paper, in addition to its descriptive nature, is how the observations relate to anything physiologically relevant. In the *in vivo* case, free tubulin concentration is likely lower, both because critical concentrations are lower in the cell and because part of the “free” tubulin pool may be engaged with other proteins. Leave along the simplification of the system and the range of condition studied, what do slow changes occurring over many hours have to do with changes in the cell of that time magnitude that do not involve changes in protein expression or composition, for example. It would also have been important, in this context, to relate the MT-MT distances in the bundles to the distances seen along MT bundles in axons.

Additional points:

- If the authors are proposing that tubulin oligomers are contributing to the cross bridges by binding to tau repeats, why will the number of cross bridges increase with more tubulin oligomers? (something they claim). I would have expected that number to be determined by amount of tau binding to MTs. The larger number would more likely be due to a redistribution of tau among fewer MTs, as the change in cross bridge numbers occurs concomitantly with MT depolymerization in their experiments. On the other hand, if tau is binding to tubulin oligomers, these are competing for tau binding to MTs. To think otherwise needs to conjure a mechanism in which both (binding to MT and tubulin oligomers) occurs

simultaneously for the same tau molecule, or that they are even coupled to one another.

- TR-SAXS over 33 hours – how was this sample not damaged? Tubulin solutions are unstable for this long without stabilizers and radiation will make the situation worse. Thus, a worry for the long-time experiments is that tubulin will start denaturing. MT are more robust, but here the interesting changes happen when the concentration of unpolymerized tubulin increases due to depolymerization. A similar fear is that protein that is seen as cross-bridges by EM, given its size and density, is actually denatured tubulin. The cross-bridges are often very large and very dense, in fact, larger and denser than the MTs. Because denaturation would be an irreversible step, an interesting experiment to debunk that doubt would be to reverse the process of compaction of the MT bundles. This could be done by reducing the amount of ions (washing with a low concentration that also contains tubulin) or, more simply, but reversing the temperature change in the temperature-induced case.

- What leads to a time evolution? Sample runs out of GTP? Protein starts to denature? Oligomers accumulate very slowly?

- The phase diagram was generated assuming that for any given ion concentration and time there are no mixtures of states (except for the intermediate helical lattice that is presented as coexisting with rings). It is not obvious to me that rings are not present together with the wide spaced lattice of MTs. Also, because the “Bragg” peaks are not obvious for 1.2 mM Ca⁺⁺ at T₀, it is hard to say that the shorter helical repeat was not present there already. For longer time frames at that concentration, there seem to be no obvious helical lattice at 6 or 12 hours, and only appears at a time when few MTs are still present.

- Interestingly, it is halfway down the paper that we are told the initial time points experience a general “relaxation from a centrifugation step”. So much is concentrated on the change in the packing of a hexagonal lattice of MTs (a lattice that they have not been able to visualize by EM), and these kind of extraneous effects are happening on the side?

- The EM images do not show the presence of helical arrangements of MTs. Also, the amount of MTs present for the two time points appears to be the opposite of what SAXS experiments indicate (there are more, not less MTs at later times). The packing of MTs at 18 hours in addition to no being hexagonal (order is missing), show an average wall to wall distance between MTs that appears smaller than the one deduced from SAXS, although there is a very large deviation from the average. In general, SAXS and EM experiments appear to show poor correspondence. This may be due to lack of ultrastructure preservation during the EM sample preparation procedures. Obviously, cryo-ET would have been a better method although much more involved. Alternatively, the assumption of a hexagonal lattice as a major contributor to the SAXS profile is simply wrong. Indeed, there is but a hint of Bragg peaks for the “wide lattice” and when the clearer intermediate lattice is obvious, MTs are already not the most abundant species.

- This statement makes no sense: “This apparent preference for 1-dimensional MT bundling, despite close lateral proximity of the linear arrays, may be due to broken cylindrical symmetry with tau distributed non-uniformly on the MT surface, consistent with reports that tau forms phase-separated complexes on the surface of MTs.” There is no relationship between a 1-dimensional array and the “phase-separated” complexes the authors are referring to.

- In Fig. 6, could they show the form factor for rings in addition to the one for MTs? Maybe in panel D? The SAXS profiles could benefit from plotting $I \times Q$ or even $I \times Q \times Q$, specially in Figure 6. What are the dash blue lines? The yellow fits show significantly more features (resolved peaks) than the experimental curves. Again, displaying $I \times Q$ or $I \times Q \times Q$ in the y axis should help better see the similarities and differences.

- What do the authors mean by “whole-mount TEM image”? This is an unusual expression. Reading the

Methods it is just the negative stain visualization of diluted samples.

- This statement also lacks sense: “despite indications of gradual MT depolymerization due to ongoing partially suppressed dynamic instability”. What is suppressing dynamic instability with time? Why will this not be compatible with gradual depolymerization?

- Where did the radius of the ring of 16.3 nm come from? In the discussion the authors talk about tau-coated tubulin rings, which by necessity would be of a heterogeneous nature. When modeling the scattering, how was this considered?

- They also state: “Enhanced cross bridging simultaneously decreases d_w while increasing the bundle domain size in the Bint state.” I would propose that they increase the order, not the size, specially when this is happening in a context of MT depolymerization

- Another statement, in this case in the discussion, that does not seem to have any firm experimental basis but be speculative is “(due to Mg^{2+} or Ca^{2+} mediated MT depolymerization of a fraction of MTs, either isolated or at the periphery of bundles where fewer cross-bridges to neighboring MTs exist)” In what data is this based? Is this a “likely” deduction?

- The authors say that the present results clarify “contradictory results” concerning the fact that “recent studies show that the MT-stabilizing drug paclitaxel, which severely reduces free tubulin, oligomers (at paclitaxel/tubulin-dimer molar ratios of $\Lambda_{paclitaxel}=1/1$), suppresses MT bundling by all six tau isoforms⁵”. There are actually TWO papers (2009 & 2017, so hardly “recent”) both of which are from the authors themselves (!), that hardly justify the claim of contradictory literature the authors refer to. Interestingly, the authors explained the effect of taxol in disrupting tau-induced MT bundling in those papers with models distinct from the one proposed here.

All new text added to the revised manuscript or Supplement is in BLUE.

Reviewer: 1

Concerns:

1.1 One major question in my mind is if the findings that are presented here from in vitro experiments are representative of what is happening in cells. Is there any evidence for this kind of ring- or oligo-based crosslinking in the axons of neurons?

The linear MT bundles found within the axon of cells are similar in structure to those found within our TEM experiments. The proposed cross linkers within our TEM images also look similar to the filamentous structures crosslinking MTs within axons (compare our TEMs Figure 6 (b,e), with published images of MT bundles within the axon (supplemental figure 1)). With this said, no tubulin-specific ring- or oligo-based crosslinking has been, to our knowledge, reported in the axon of neurons. We note however that MTs in the axon-initial-segment are reported to be unusually dynamic[Zempel *et al*, 2017], so we expect that free tubulin oligomers are available where linear MT bundles are observed. And while the composition of the cross-bridges observed in cells is not fully known, we do not assert that this is the primary mechanism for MT bundling within cells, as the cell is a much more complex system. We do, however, want to highlight that reconstituted in-vitro experiments containing just tau, tubulin, and GTP, are able to produce “linear bundles” which closely mimic those found in the axon, and that the availability of free tubulin appears to have a strong effect on the architecture of these in-vitro bundles.

1.2 How does the bundling observed here relate to microtubule organization by neurofilaments in axons?

In the axon-initial-segment (AIS), the linear/branched MT bundles immersed within the neurofilament network appear to be phase separated from the neurofilament network. Nevertheless, the phase separated MT bundles in axons appear to be stabilized by attractive forces between the MTs (i.e. where linkages between MTs are clearly visible in end-views of cross-sections of axons). Our cell free studies show that MT bundles, even in the absence of the neurofilament network, possess similar linear architectures to those in axons and are stabilized by attractive forces due to the tubulin oligomer/tau intervening network.

1.3 Could the authors discuss a bit more how they think the rather disordered crosslinking by tau and tubulin oligos and rings leads to a preferred distance between the microtubules as observed

in SAXS? I could imagine smaller spacing with short oligos or large spacing with a clump of oligos or rings in between.

The SAXS peaks are measuring a weighted average of the spacings between the MTs (which includes the shorter and longer tubulin oligomers coated with tau). However, while the tether length that connects two MTs can vary by multiple nanometers (due to variations in size, conformation, and orientation of the disordered crosslinker), we expect the measured lattice constant through SAXS to be dominated by the diameter of the tubulin ring, as the tubulin ring's curvature is well defined (i.e. generates a constant spacing) and requires the least number of tau-tubulin interactions to form a bridge (i.e. is likely to occur). Exactly as you suggest, we too suspect that the smaller spacing of the B_{int} state is a result of the shorter average tubulin oligomer size (as seen through SAXS) surrounding still polymerized MTs.

1.4 Microtubule depolymerization is a physiological part of dynamic instability, but can also occur through protease activity or other degradation. During tubulin preparation from tissue, the whole purification procedure uses selection of the polymerizable fraction from all the rest, but that selection is of course never perfect. How reproducible were the results using different batches of tubulin from different preparations?

For a given concentration, two samples prepared from the same batch of tau and tubulin showed similar stability for wide-spacing bundles, with the time of transition differing by ~30 minutes. Comparing the time of transition from two samples at the same cation concentration but from different tubulin preparations showed a difference in time of transition by multiple hours. Despite this, the overall trend of each phase diagram (increased cation concentration decreases the time of stability for the B_{WS} state) was the same across several batches of tubulin purified both from bovine and porcine brains (though we only present bovine data here). An example of the reproducibility across different tau and tubulin batches has been added to the SI of the paper and is shown in the phase diagram below.

Text added to manuscript (in section entitled: Time-dependent SAXS generates kinetic phase diagrams revealing three distinct assembly structures for ab-tubulin/tau/divalent cation/GTP mixtures at 37°C):

We also note that while the time of transition to the intermediate state was reproducible when comparing samples from the same batch of tau and tubulin, variability was observed when comparing the time of transition across two different batches (Supp. Fig. 4). Because of this, all time-based experiments presented here were conducted using the same batch of tau and tubulin.

Figure and corresponding caption added to the Supplemental Section (New Fig S4):

Comparison of kinetic phase diagrams for two different batches of tau and tubulin. The pink, yellow, and green colored regions represent the region of ring, intermediate, and wide-spacing states from the tau and tubulin batch plotted in Fig. 3 of the main text. The magenta circles, yellow triangles, and green squares however show the identified phase for samples prepared from a different batch of tau and tubulin. Comparison of a given marker color with its surrounding background color shows whether the two samples (from different tau and tubulin batches) were in the same phase. As you can see, the general trend that increasing cation concentration decreases BWS stability is observed regardless of the tau/tubulin batch. However, while both batches show the same trend, when comparing the two, the second batch (signified with markers) transitioned multiple hours sooner. Lastly, we emphasize that all data presented in the paper with exception to the temperature experiments were done with the same batch of tau and tubulin.

1.5 In the time dependent SAXS experiments, is there any danger of prolonged X-ray exposure causing damage in the samples or were the samples not exposed repeatedly?

Samples were exposed to X-ray radiation for one second, once every 3 hours over a 33-hour period. Thus, *the total radiation exposure of a sample was 12 seconds over a period of 33 hours*. The three-hour delay between 1 second exposures when combined with the fact that our samples are both regenerative and in solution causes the effective dosage from repeat exposures to be very low.

To measure the effects of prolonged radiation exposure to WS bundles, a sample without added salt was exposed to 60 one-second exposures over 2 minutes (i.e., 1 second on, 1 second off). Plotted below are the results of the experiment. As expected, prolonged X-ray exposure does result in denatured tubulin, where the scattering intensity at very low q is seen to increase. However, no visible radiation damage was noted for the first 5 seconds of exposure. We stress that the observed transition from the wide spacing to the intermediate spacing bundles may occur as early as the first (i.e. “0” time), second, or third exposures depending on the divalent ion concentration (e.g. see Figure 3A).

Furthermore, despite massive reduction in MT-polymerized tubulin over 60 seconds of irradiation (likely due to denatured tubulin giving rise to enhanced SAXS at very small q), the intermediate state was not observed (i.e. d_w - w remains constant and Bragg peaks broaden) demonstrating radiation damage does not produce the phase transition to the intermediate state. We also note that the fitted scattering parameters for the (radiation damaged) free-tubulin in the sample below are very different from those measured from experiments within the paper and further suggest that the effects of radiation damage were nominal within our experiments.

Text added to manuscript (in section entitled: Time-dependent SAXS generates kinetic phase diagrams revealing three distinct assembly structures for $\alpha\beta$ -tubulin/tau/divalent cation/GTP mixtures at 37°C).

Samples were exposed to 1 second of synchrotron radiation once every 3 hours for a 33-hour period.

We note that the observed depolymerization is not occurring due to denaturation from multiple synchrotron exposures and that separate experiments testing the effects of prolonged x-ray radiation were unable to produce the Bint state (Supp. Fig. 3).

Figure added to the Supplemental Section (New Fig. S3):

Time-dependent SAXS data shows that radiation damage produces distinct scattering features not observed in experimental conditions. (Left) Raw SAXS data was collected using 1-second exposures every two seconds for a period of two minutes. At initial and early timepoints (purple to blue curves), features of the B_{ws} state are prevalent, whereas scattering from later exposures (green to red curves) show increased scattering at low q values and decreased scattering from bundled MTs. Inset shows the first 11 exposures and highlights the continuous increase in scattering at q values associated with minima from the MT form factor. (Right) Zoomed-in raw SAXS data (from Figure 7) from samples containing 1.8 mM and 0.6 mM Mg^{2+} (top and bottom, respectively) show the distinct features of samples that do and do not transition from the B_{ws} state to the B_{int} state, respectively. Specifically, scattering at MT form factor minima does not increase while the B_{ws} is the dominant structural state, but scattering at these form factor minima increase sharply upon transition to the B_{int} state. Following the transition to the B_{int} state, further increases in scattering at FF minima is minimal. Data from Figure 7 was collected as described throughout the main text, with 1-second exposures every 3 hours. The distinguishing features of the intentionally irradiated sample in A are not observed in the experiments reported in the main text.

Reviewer: 2

Concerns:

2.1 Data clearly shows the Bws transitions to the Bint and rings upon addition of divalent ions, temperature decrease, and time progress. However, it is hard for me to accept the conclusion that the Bint requires free tubulins. Figure 4 shows that the first minima goes up, based on which the author's claims the formation of rings, but the Bint doesn't form in this case. So, doesn't the increase of the minima in this case indicate the increase of free tubulins?

As outlined in the text, while increased scattering at only the first form factor minimum does indicate an increase of tubulin mass not in the MT lattice; we know this tubulin cannot be in the tubulin ring conformation as that would require a rise in scattering at *all minima* determined by the ring's form factor. Plotting figure 4 of the manuscript without offset (below) and looking at the Bessel function minima, the first minimum (at $q \sim 0.2 \text{ \AA}^{-1}$) does increase with increased monovalent concentration, however all other minima—aside from the sample with 150 mM added KCl—do not. Thus, the fitting shows no measurable increase in tubulin ring concentration and smaller tubulin oligomers for the 25 mM through 125 mM samples.

Regarding the 150 mM sample, we note that earlier studies have shown that the diameter of MTs decrease with decreasing tau coverage on MTs [Choi *et al*, 2009] consistent with our finding shown in supplemental figure 4. Furthermore, it is also known that tau binding is salt-dependent with increased KCl concentrations decreasing tau binding [Choi *et al*, 2009]. Thus, the observed shift in the Bessel function minima to higher q for the 150 mM sample in Figure 4 is consistent with the fact that there is dramatically less tau bound to the MT surface for this sample (fitted MT radius values for the 150 mM KCl scattering profile are similar to samples prepared at a 1:80 tau to tubulin ratio without added salt, see supplemental figure 4). Resulting fitting parameters for the 150 mM KCl scattering profile do indicate a higher amount of tubulin rings and oligomers when compared to the samples at the lower salt concentrations (25 through 125 mM added KCl), however, the tau-tubulin interaction strength is approximately 4x weaker in these conditions, so we expect a smaller contribution to the intervening network by free tubulin that is less able to form complexes with tau.

2.2 They claim that GTP concentration or depletion is not a factor, then what happens if they use a much smaller amount of GTP? or why not present data with less GTP?

We thank the reviewer for this important point. In the paper our claim is that the transition between the wide-spacing and intermediate spacing bundles occurs in the presence of GTP. We do not aim to claim that the GTP concentration has no role in the observed time delay of the divalent ion-induced intermediate state, but that the divalent ion-induced MT depolymerization observed in samples prepared at 2 mM GTP is happening in a GTP rich environment and the differing time delays are primarily a result of the different concentration of added divalent cations.

Indeed, data added in the revised manuscript shows that lowering (or increasing) the GTP concentration effectively shifts the kinetic phase diagram in that it reduces (or increases) the time of stability for wide-spaced MT bundles at a given concentration compared to the intermediate MT bundle state. This is consistent with the expectation that GTP enhances MT polymerization while divalent ions have the opposite effect and tend to enhance MT depolymerization.

Text added to manuscript (in section entitled: Time-dependent SAXS generates kinetic phase diagrams revealing three distinct assembly structures for ab-tubulin/tau/divalent cation/GTP mixtures at 37°C).

Increasing (or decreasing) the initial GTP concentration for a prepared sample also effectively shifts the kinetic phase diagrams shown in figure 3 by either increasing (or decreasing) the time delay before the intermediate phase transition, and by

lowering (or increasing) the minimum divalent ion concentration necessary for the intermediate phase to be observed by 33 hours. Figure 5 shows scattering profiles for samples prepared with 1.5 mM added Mg (Fig. 5A) and Ca (Fig. 5B) at time t_0 and at varying GTP concentrations. Data shows clear examples of all three labeled tubulin phases with varying GTP concentrations and highlights that the concentration necessary to induce the intermediate phase is lower for Ca compared to Mg.

While the time delay of the intermediate phase transition can be modulated with GTP concentration, it is important to note that the time dependency shown in Figure 3 is primarily driven by the increased concentration in divalent cations and not due to GTP within our system being depleted.

New figure and caption added to manuscript (New Fig. 5):

Synchrotron SAXS data reveals that the tubulin-tau assembly state is GTP dependent.

SAXS data (open circles) and fit lines (solid colored lines) of tubulin/tau/GTP mixtures at 37 °C and 4RL-tau to tubulin-dimer molar ratio $\Phi_{4RL} = 0.05$ with increasing GTP concentration and 1.5 mM $MgCl_2$ (A) or $CaCl_2$ (B). SAXS scans are offset for clarity, and fit lines are color-coded to the dominant scattering feature for each data point.

2.3 The authors did not put all the parameters obtained from the fit. Obviously, data of rings shows much finer oscillations and shifts of minima toward smaller q regions, suggesting a larger ring size. And the background increases more at smaller angles, for example, at the first minima and lower q . Is this suggesting the formation of a larger object instead of free tubulins?

Yes, the gradual increase in scattering over time at very low q does imply the formation of large scale tubulin structures. We believe this is from a mixture of aggregated free tubulin, aggregated tubulin rings, and denatured tubulin from X-ray radiation damage. We emphasize that an increase in scattering over time at low- q , though common, wasn't observed for all samples, and that no correlation was observed between the formation of these larger scale structures with the onset of the intermediate phase transition (whereas the increased prevalence of smaller tubulin oligomers and tubulin rings always coincided with a decrease in d_{w-w}). Lastly, we note that these structures may also crosslink neighboring MTs or MT bundles. This, however, still makes sense within our model as their large, amorphous, and polydisperse structure can't mediate MT bundling with a homogenous wall to wall spacing and thus wouldn't produce discernable Bragg peaks through SAXS.

2.4 Figure 7 may indicate that the SAXS invariant may stay the same, independent on time. If bundles become free tubulins and there's no loss of tubulins, this looks right in the invariant point of view. But, data shows increase of larger structures instead of increase of scatterings from smaller objects, for example, free tubulins dimers. How can this be explained?

The experiments plotted in figure 7 were conducted over 33 hours. Over this time, MTs are slowly depolymerizing with the mass of free tubulin increasing as GTP hydrolysis is ongoing (i.e. before GTP depletion). The slow creation of free tubulin oligomers coated with tau, over time, will create larger structures of aggregated tubulin oligomers that give rise to the enhanced SAXS observed at very small q . We note, however, because the MT depolymerization is very slow in the wide spacing state at low divalent ion concentrations, the scattering from the free tubulin relative to the MT bundles is essentially below the detection level.

2.5 In biomolecules/biopolymers, upon the addition of divalent ions, the sequence of structural transitions, free particles/chains-crystalline assemblies-free particles/chain, is commonly observed. Isn't the author's observation consistent with those?

In the absence of a reference, we are unfortunately unable to properly respond to this question.

2. 6 Why is the ionic effect much slower than the temperature effect? The free tubulin formation may be an important factor, but there needs to be some explanation about this kinetic effect. According to Ca^{2+} data from Figure 3A, tubulin rings may form at even low Ca^{2+} concentration by just waiting for long enough.

For ion induced depolymerization, depolymerization occurs from the disruption of the lateral bond due to the divalent cation between the M-loop on one β -tubulin and the H1-S2-loop

on the neighboring β -tubulin [Ojeda-Lopez *et al*, 2014]. The depolymerization occurs if a long enough section of neighboring protofilaments (of order the persistence length of protofilaments) are unzipped simultaneously and thus is a long time event. Cold temperature on the other hand renders GTP tubulin assembly-incompetent due to an induced longitudinal conformational change in GTP-tubulin over the entire protofilament. As a result, the intermediate state is expected to occur on faster timescales with temperature (where we find it to be of order of minutes) compared to with divalent ions (where we find it to be of order hours).

2.7 Please, do not omit experimental details. For example, what is t_0 ? The authors define it as a start of measurement, which seems too vague. It should be something like the time of mixing divalent ions. What are the plastics used for embedding? What are used for the A_{hk} in equation 2 and A_{ring} in equation 4?

The methods for our plastic embedding process have been elaborated upon and now reads:

Pellets were then embedded in an epoxy-based low viscosity embedding media prepared by mixing 5 g of ERL 4221 (3,4 Epoxy Cyclohexyl Methyl 3,4 epoxy Cyclohexyl Carboxylate), 4 g of D.E.R. 736 (diglycidyl ether of propylene glycol), 13 g of NSA (nonyl succinic anhydride), 0.2 g of EASE (PolyCut-Ease), and 0.2 g of DMAE (diethylaminoethanol). Dehydrated sample pellets were infiltrated with embedding media, poured into flat embedding molds, held at 65 °C for 48 hours to polymerize, and then cooled overnight prior to sectioning.

A_{hk} and A_{ring} are scattering amplitude values that are proportional to the number of tubulin molecules within the B_{MT} and ring states. Their values however are in arbitrary units and were omitted from the text as their results are highly sample- and source-dependent. Normalized values for A_{ring} however are useful and are plotted in figure 8. Non-normalized values ranged between $0-1 \times 10^{-10}$ and $0-7 \times 10^{-8}$ for A_{10} and A_{ring} respectively.

The time t_0 corresponds to the time taken for sample preparation plus the time to get the sample on the x-ray diffractometer for the first measurement (t_0 can be approximated to be an hour and fifteen minutes after mixing protein and cations on ice, which includes a 30-minute incubation period and a 30-minute centrifugation period as described in Methods). While t_0 can vary between samples by a couple minutes, the significant structural changes we are seeing are on order several hours. We have added text to the paper to clearly define t_0 when first referenced. We respectfully disagree with the reviewer's suggestion to change our definition of t_0 but feel our clarified definition suffices.

Text added to manuscript (in section entitled: Time-dependent SAXS generates kinetic phase diagrams revealing three distinct assembly structures for ab-tubulin/tau/divalent cation/GTP mixtures at 37°C).

Reactions underwent 30 minutes of polymerization at 37 C, followed by 30 minutes of centrifugation at 37 C and then loaded onto the x-ray diffractometer with initial data points taken approximately 15 minutes after centrifugation. Samples were exposed to 1 second of synchrotron radiation once every 3 hours for a 33-hour period. Analysis of all azimuthally averaged SAXS profiles reveals three distinct concentration-dependent tau-tubulin structural phases outlined in Fig. 2A.

At initial timepoint t_0 (where t_0 signifies the initial timepoint for data collection), all samples below a threshold divalent concentration of 1.4 mM added Ca^{2+} or 2.4 mM added Mg^{2+} exhibited MT bundling characteristics indistinguishable from controls with no added divalent cations (Supp. Fig. 2).

Reviewer: 3

Concerns:

3.1 There are ways in which the authors could more directly test this hypothesis. For example, if soluble GDP-bound tubulin were to be added, would that speed up the formation of the more compact bundles? Because of the nature of their experiments, MT needs to depolymerize to get the excess rings that the authors claim causes the tightening of the MT arrays, but ultimately this leads to full depolymerization of the MTs. If all that is needed is extra tubulin in an unpolymerized state to cause the tightening of the bundles, then adding GDP tubulin to the MT-tau mixture under conditions that promote wider spacing should lead to their compaction without disassembly. Their temperature experiment still relies on a depolymerization process that ends with the disappearance of the MTs. Additionally, temperature can have many effects beyond MT depolymerization, for example, on the dynamics/conformation of tau.

In our experimental system, the transition between the widely-spaced MT bundle state and the more compact intermediate MT bundle state involves a rapid proliferation of tau-coated tubulin rings (due to partial MT depolymerization of MTs at random positions *within the bundles*) where the newly formed tau-tubulin rings, that lead to the more compact bundle state, are created from MTs distributed within the bundles. The suggestion by the reviewer of adding GDP-bound tubulin to an already existing widely-spaced bundle state to achieve the more compact state has the following complications associated with it. The biggest being that there is a large kinetic and entropic barrier to mixing the proposed soluble, GDP-tubulin with a dense, centrifuged, *phase separated* protein network. Both tubulin oligomers and MTs are overall negatively charged and thus electrostatically repel each other. Short range tau-tubulin interactions are required for the connection of free tubulin between MTs, however these interactions also compete with tau connecting neighboring tubulin rings outside of the widely spaced bundles, creating large aggregates that tend to phase separate from the bundles. The size of both the proposed added GDP-tubulin structures and the tubulin structures within our bundled system cause the entropy of mixing to be extremely low. Lastly, the availability of excess GTP in our system, without which depolymerization would occur, will freely exchange into the proposed GDP-bound tubulin oligomers added to the system.

Alternatively, as suggested by the reviewer, we performed the temperature reversibility experiments, which were successful, and the *new data have been incorporated in the revised manuscript (this is discussed in concern 3.8 below)*. It shows that the bundle transition is fully reversible between 37°C (with bundles in the widely spaced state) and 21°C (where bundles transition to the more compact intermediate state due to temperature-induced ring formation (from MT depolymerization of MTs at random positions *within the bundles*)). This implies that when the system is brought from 21 °C back up to 37°C, the tubulin oligomers re-polymerize into the MT lattice, and, as this is occurring, the samples re-transition into the wide-spacing state.

Thus, the temperature-reversibility data are consistent with our model that adding additional curved tubulin oligomers drives the transition to the intermediate state and removing oligomers reverts the system back to the wide-spacing state.

3.2 The authors state: “we expect these curved tubulin structures to be coated with tau” (line 185-6). Indeed, it is an “expectation” of the authors, without a solid demonstration. The interpretation of the SAXS data and EM in terms of cross-bridges of tubulin oligomers-bound tau needs corroboration that is more direct. As it exists, the SAXS is not sufficient to support this proposal by itself, even if it may be compatible with it, and the EM shows densities that are more easily explained by aggregated protein than tubulin oligomers/rings. The model may have some appeal, but it is in no way proven by the data presented.

Tau has been previously shown to increase the average MT diameter in a concentration dependent manner [Choi *et al*, 2009]. This was also observed within our data where the MT Bessel function minima shifts to lower q as the tau to tubulin ratio increases, (SI Figure S6, plotted below to the left, zoomed in and with different vertical offset for visual clarity). Throughout all time based SAXS experiments where MT bundles were observed to transition from the wide-spacing to intermediate state, the measured radius of the MT never increased but instead either remained constant or slightly decreased in size. This strongly indicates that the density of tau bound to a MT's surface is roughly constant throughout depolymerization and implies that tau is remaining bound to the depolymerized tubulin oligomers (i.e. if the depolymerizing tubulin oligomers were not coated with tau then that would imply an increase in the tau coating the MTs and thus a change in the MT diameter which does not occur). Reports have shown both that tau binds to soluble tubulin [Li and Rhoades, 2017; Li *et al*, 2015] and that tau preferentially binds to curved over straight tubulin structures [Duan *et al*, 2017], further validating the assumption that depolymerization products are coated with tau.

3.3 The authors state “TEM data provides direct evidence that tubulin oligomers...”, but that is not true. What the extra density is, multiple tau molecules, tubulin oligomers or tubulin/tau aggregates of denatured material, it is not possible to say. The EM images provided are, by their nature, very low resolution. In this day and age, the authors should find ways to pursue imaging using cryo-EM, or more specifically, cryo-electron tomography. Their new model is radical enough to require the test of serious proof.

The TEM images that we show use the involved method of “plastic embedded” imaging. In this method the shapes of objects are essentially not altered (see further discussion in section 3.12 of our response). This is in contrast to the whole-mount TEM method in vacuum where object shapes are often distorted.

We agree with the reviewer that cryo-TEM is a more accurate method of obtaining quantitative length scales of objects (which is what we do using SAXS instead). However, cryoTEM would not be able to image single strands of intrinsically disordered tau either on MTs or on tubulin oligomers (i.e. cryo-TEM works well for imaging of tau fibers which have secondary structures).

In response to the reviewer, we changed the wording of the sentence from “provides direct evidence...” to “suggests...” in order to be more accurate. It can be said with relative certainty however that the extra density is a combination of tau and tubulin. In addition to the reasons provided within the text, it is unlikely that the observed dense connections are from multiple tau molecules as studies have shown that crowding agents are required for pure tau-tau coacervates to form at our sample’s ionic concentration [Kanaan *et al.* 2020]. Furthermore, it is highly unlikely that multiple tau molecules would happen to form ring structures at the same diameter as tubulin rings (which are pointed to in the figure), thus the observed density likely

contains tubulin. Tubulin oligomers and tubulin rings however are unable to stick to the walls of MTs alone, so the visualized cross-bridges can not be composed of just tubulin oligomers. Thus, through elimination, the observed densities crosslinking MTs are most likely to contain both tau and tubulin.

We point out that through hundreds of EMs images the extra density is not randomly/uniformly distributed throughout the sample and is generally only found connected to or surrounding polymerized MTs. Thus, the likelihood of the extra densities being aggregates of denatured tubulin is low. We also observed this density to go down with time through EMs monitoring the first hour of polymerization (implying that the observed tubulin structures aren't denatured and are being polymerized into the MT over time). Lastly we have added two additional experiments to the paper that show 1) MT depolymerization through radiation damage does not induce the intermediate state, meaning the effects reported are not due to denatured material, and 2) reversing the temperature from 21 C to 37 C brings intermediate spaced bundles back into the wide spacing along side previously depolymerized tubulin re-polymerizing into the MT lattice.

3.4 The authors state that the MT spacing in bundles is much larger than the tau PD, but fully extended PD and PRD will be able to cover those distances. They provide information for experimental radius of gyration for tau molecules under different conditions that invokes a significant compaction of an extended chain to about 40 Å (this is approximately the size of a tubulin monomer, when just one pseudo-repeat of tau extends over two when bound to a MT). That length is not compatible with binding to tubulin oligomers/rings. I do not see how the authors can use those numbers to justify that the distances between MTs cannot be explained by tau by itself, but then assume that those numbers are compatible with binding to multiple tubulins in oligomers and to MTs simultaneously! Have they obtained data about the radius of gyration of tau in the presence of tubulin rings alone, say, generated by GDP tubulin at low temperatures? Are those more compatible with the MT-MT distances they observe?

The MTBR of tau contains specific binding sites for tubulin. Thus while the MTBR of tau is significantly stretched when binding to tubulin, this conformation is still favorable as its free energy decreases. Most importantly however, tubulin has a defined structure which forces tau to remain stretched upon binding. Tau-Tau binding sites of similar binding enthalpy within the PD of tau have not been reported. Moreover, if such binding sites did exist, it still wouldn't explain why tau, an intrinsically disordered protein, would remain stretched after tau-tau dimerization. Because of this we say it is unlikely that tau would naturally stabilize MT bundles at a dw-w ~45 nm (Fig. 8A and C) given the tau:tubulin ratio and monovalent salt concentration with a very short Debye length (of order 9 Å). It is however close to the outer diameter of a tubulin ring (40 nm), which is what you would expect if tubulin oligomers helped mediate MT bundling.

3.5 Generally, the paper is very descriptive and lacks molecular mechanistic insights. Is the compaction affecting the number of repeats that are attaching to MTs per tau molecule? The amount of oligomers bound to tau? What is the molecular rate limiting step that makes the evolution with time so slow?

We believe the compaction is due to the increased concentration of tau-coated tubulin rings and small tubulin oligomers. In the wide-spacing state, MT polymerization is preferred and the majority of tubulin is polymerized into the MT lattice. Because of this, the probability of tau-tubulin cross-bridges is lower, and the dominant spacing observed through SAXS is close to the diameter of a tubulin ring (i.e. the curvature of tau bound tubulin oligomers) as it has a well defined spacing and requires the least number of tubulin-tau-tubulin bridges. During massive MT depolymerization events, however, the influx of smaller tubulin oligomers increases the probability of more direct MT bridging and thus lowers the average distance between neighboring MTs (and lowering the observed dw-w). We have added a few sentences to the paper making our model for compaction more clear to the reader.

As pointed out in 3.2, SAXS does unambiguously show that the density of tau bound to a MT's surface is approximately constant throughout the transition between the MT bundled states. Beyond that, the molecular details of our model are descriptive as further details are beyond the capabilities of SAXS and beyond the scope of this paper. The paper's goal is to show that MT bundling architecture is strongly linked to the surrounding free tubulin oligomer concentration which (especially with the addition of more data) we feel we have adequately provided evidence for. This, tied to the evidence of tau-tubulin cross-bridges in TEM, leads us to propose that both tau and tubulin play a role in stabilizing in-vitro MT bundles.

3.6 Another problem with this paper, in addition to its descriptive nature, is how the observations relate to anything physiologically relevant. In the in vivo case, free tubulin concentration is likely lower, both because critical concentrations are lower in the cell and because part of the "free" tubulin pool may be engaged with other proteins. Leave alone the simplification of the system and the range of conditions studied, what do slow changes occurring over many hours have to do with changes in the cell of that time magnitude that do not involve changes in protein expression or composition, for example. It would also have been important, in this context, to relate the MT-MT distances in the bundles to the distances seen along MT bundles in axons.

Since MTs are undergoing dynamic instability in the axon-initial-segment, we expect that free tubulin oligomers are available where linear MT bundles are observed. We highlight that our reconstituted in-vitro experiments containing just tau, tubulin, and GTP, are able to produce "linear bundles" which closely mimic those found in the axon-initial-segment, and that the availability of free tubulin appears to have a strong effect on the architecture of these in-vitro bundles. The proposed cross linkers within our TEM images also look similar to the filamentous

structures crosslinking MTs within axons (compare our TEMs Figure 6 (b,e), with published images of MT bundles within the axon (supplemental figure 1)).

We have added to the text a sentence relating the spacings we see to those found for MT bundles in the axon-initial-segment (AIS). Sentences have also been added to make it clear that we do not attribute any physiological significance to the exact time in which a given sample transitioned into the intermediate state. The major significance of the paper is that MT depolymerization appears to also affect the architecture of in-vitro MT bundles. Dynamic instability is a critical characteristic of MT functionality and thus our findings have potential importance to the general cell-biology community.

3.7 If the authors are proposing that tubulin oligomers are contributing to the cross bridges by binding to tau repeats, why will the number of cross bridges increase with more tubulin oligomers? (something they claim). I would have expected that number to be determined by the amount of tau binding to MTs. The larger number would more likely be due to a redistribution of tau among fewer MTs, as the change in cross bridge numbers occurs concomitantly with MT depolymerization in their experiments. On the other hand, if tau is binding to tubulin oligomers, these are competing for tau binding to MTs. To think otherwise needs to conjure a mechanism in which both (binding to MT and tubulin oligomers) occur simultaneously for the same tau molecule, or that they are even coupled to one another.

As discussed in concern 3.2, our data and previous reports both validate the assumption that tau is likely bound to tubulin oligomers following MT depolymerization. To address the need for a mechanism by which tau simultaneously binds multiple oligomeric or MT-bound tubulins, we added a reference which explains this. With these in mind, it's straightforward that increasing the number of potential cross-links (i.e. tau-coated oligomeric tubulin) will increase the amount of cross-linking.

You are correct in assuming that increasing the amount of tau bound to MTs also increases the number of cross-bridges. SI figure 6 clearly shows that increasing the tau:tubulin ratio from 1:60 to 1:5 dramatically increases effective MT bundle size. However, as discussed earlier, if the tau were redistributed across the MT upon depolymerization (rather than remaining bound to depolymerizing tubulin) this increased tau density would also result in an increased MT diameter during the intermediate state, which is not observed. Our data instead suggests that the density of tau bound to MTs during depolymerization is constant or slightly decreasing. Moreover, measured d_{w-w} in SI figure 6 shows that increasing the amount of tau bound to MTs doesn't decrease d_{w-w} but either remains constant or increases at very high tau:tubulin ratios. Thus, our results are incompatible with your proposed model, and increases in tau-coverage on the MT surface can not explain the different bundling architecture of the B_{int} state.

3.8 TR-SAXS over 33 hours – how was this sample not damaged? Tubulin solutions are unstable for this long without stabilizers and radiation will make the situation worse. Thus, a worry for the long-time experiments is that tubulin will start denaturing. MT are more robust, but here the interesting changes happen when the concentration of unpolymerized tubulin increases due to depolymerization. A similar fear is that protein that is seen as cross-bridges by EM, given its size and density, is actually denatured tubulin. The cross-bridges are often very large and very dense, in fact, larger and denser than the MTs. Because denaturation would be an irreversible step, an interesting experiment to debunk that doubt would be to reverse the process of compaction of the MT bundles. This could be done by reducing the amount of ions (washing with a low concentration that also contains tubulin) or, more simply, reversing the temperature change in the temperature-induced case.

Regarding the “reversibility experiment” concern:

Following the reviewer’s suggestion we have successfully performed the reversibility experiment. The revised text includes the temperature reversibility experiments. Based on separate experiments testing the rate of MT depolymerization with temperature, we observed that MTs for our system begin to depolymerize more rapidly at temperatures below 21.5 °C—with increased depolymerization rates at colder temperatures. The figure added to the text is of a sample with 2.0 mM of added Mg^{2+} . Every 10 minutes a 1 second exposure of the sample was taken. For the first 30 minutes the temperature of the sample was at 37 °C, the sample was then cooled to 21 °C for 30 minutes, and then the temperature was restored to 37 °C. As you can see, dropping the temperature clearly induced both MT depolymerization as well as the B_{int} architecture. Moreover, reversing the temperature after 30 minutes to back to 37 °C shows clear reversibility of the depolymerized tubulin scattering contribution and increase in B_{MT} scattering. Along with the reduction in depolymerized tubulin mass there is a corresponding increase in d_{w-w} back to wide-spacing values and reduction in the effective bundle domain size.

To summarize the new reversibility data: The data shows that the bundle transition is fully reversible between 37°C (with bundles in the widely spaced state) and 21°C (where bundles transition to the more compact intermediate state due to temperature-induced ring formation from MT depolymerization of MTs at random positions *within the bundles*). This implies that when the system is brought from 21 °C back up to 37°C, the tubulin oligomers re-polymerize into the MT lattice, and, as this is occurring, the samples re-transition into the wide-spacing state. This is consistent with our model that adding additional curved tubulin oligomers drives the transition to the intermediate state and removing oligomers reverts the system back to the wide-spacing state.

Text added to manuscript (in section entitled: Increasing unpolymerized tubulin oligomer content induces the transition from the wide-spacing (B_{ws}) to the intermediate (B_{int}) bundled microtubule state).

A sample polymerized with 2.0 mM added Mg^{2+} was monitored while the temperature was cycled 30-minute periods at 37 °C, 21 °C and 37 °C again. SAXS data from this temperature cycling experiment (Figure 9) revealed that the otherwise stable wide-spacing state rapidly transitioned to the B_{int} state immediately following the drop in temperature to 21 °C. The characteristic shift in hexagonal Bragg peaks to higher q was observed, coinciding with a sudden increase in tubulin ring scattering. At 21 °C, $d_{\text{w-w}}$ decreased to a minimum value of 21.8 nm (change in wall-to-wall distance, $\Delta d_{\text{w-w}} = 8.5$ nm), and scattering from tubulin rings increased over the 30-minute incubation at 21 °C. Increasing the temperature back up to 37 °C following the 30-minute incubation at 21 °C, the microtubule bundles reverted to the wide-spacing state, as indicated by the increase in $d_{\text{w-w}}$ to 30.5 nm, decrease in the coherent bundle size, abrupt decrease in scattering from tubulin rings, and corresponding increase in B_{MT} scattering. This result further demonstrates the correlation between free tubulin oligomer content and MT bundle architecture as the system transitions between the B_{ws} and the B_{int} states.

New figure added to the manuscript (New Fig. 9):

Time-dependent synchrotron SAXS reveals that the microtubule phase transition can be induced and reversed by cycling temperature from 37 to 21 and back to 37 °C (A) SAXS data (open circles) and corresponding fits (solid lines) for a sample prepared with 2.0 mM MgCl_2 added to standard PIPES buffer at pH 6.8. The sample was held at 37 °C for 30 minutes, quickly reduced to 21 °C and held for 30 minutes, then quickly cycled back up to 37 °C. Data was taken

every 10 minutes. Fit lines are color-coded for the wide-spacing state (37 °C, green) and the intermediate state (21 °C, yellow).

Regarding the radiation damage part of the concern:

Samples were only exposed to X-ray radiation for one second once every 3 hours unless otherwise noted. Thus, *the total radiation exposure of a sample was 12 seconds over 33 hours*. We have added text to the paper to avoid the possible confusion that samples were exposed to synchrotron radiation for a continuous 33 hours. We note that the three hour delay between 1 second exposures when combined with the fact that our samples are in solution and regenerative causes the effective dosage from repeat exposures to be very low. We have also conducted separate experiments measuring the effects of radiation damage to our samples and have added the results to the supplement (and shown below).

The graph below shows a B_{WS} sample prepared without added salt and exposed to 60 one-second exposures over 2 minutes (i.e., 1 second on, 1 second off). As expected, prolonged X-ray exposure does result in a loss of MT mass with a corresponding increase in denatured tubulin—where scattering at very small q is seen to increase. However, no visible radiation damage was noted for the first 5 seconds of exposure. We stress that the observed transition from the wide spacing to the intermediate spacing bundles may occur as early as the first (i.e. “0” time), second, or third exposures depending on the divalent ion concentration (e.g. see Figure 3A).

Furthermore, despite massive reduction in MT-polymerized tubulin over 60 seconds of irradiation (likely due to denatured tubulin giving rise to enhanced SAXS at very small q), the intermediate state was not observed (i.e. d_w - w remains constant and Bragg peaks broaden) demonstrating radiation damage can not produce our results. We also note that the fitted scattering parameters for the (radiation damaged) free-tubulin in the sample below are very different from those measured from experiments within the paper and further suggest that the effects of radiation damage were nominal within our experiments.

Text added to manuscript (in section entitled: Time-dependent SAXS generates kinetic phase diagrams revealing three distinct assembly structures for ab-tubulin/tau/divalent cation/GTP mixtures at 37°C).

Samples were exposed to 1 second of synchrotron radiation once every 3 hours for 33 hours.

We note that the observed depolymerization is not occurring due to denaturation from multiple synchrotron exposures and that separate experiments testing the effects of prolonged x-ray radiation were unable to produce the intermediate state (Supp. Fig. 3).

Figure and corresponding caption added to the Supplemental Section (New Fig. S3):

Time-dependent SAXS data shows that radiation damage produces distinct scattering features not observed in experimental conditions. (Left) Raw SAXS data was collected using 1-second exposures every two seconds for a period of two minutes. At initial and early timepoints (purple to blue curves), features of the B_{ws} state are prevalent, whereas scattering from later exposures (green to red curves) show increased scattering at low q values and decreased scattering from bundled MTs. Inset shows the first 11 exposures and highlights the continuous increase in scattering at q values associated with minima from the MT form factor. (Right) Zoomed-in raw SAXS data (from Figure 7) from samples containing 1.8 mM and 0.6 mM Mg^{2+} (top and bottom, respectively) show the distinct features of samples that do and do not transition from the B_{ws} state to the B_{int} state, respectively. Specifically, scattering at MT form factor minima does not increase while the B_{ws} is the dominant structural state, but scattering at these form factor minima increase sharply upon transition to the B_{int} state. Following the transition to the B_{int} state, further increases in scattering at FF minima is minimal. Data from Figure 7 was collected as described throughout the main text, with 1-second exposures every 3 hours. The distinguishing features of the intentionally irradiated sample in A are not observed in the experiments reported in the main text.

3.9 What leads to a time evolution? Sample runs out of GTP? Protein starts to denature? Oligomers accumulate very slowly?

For divalent cation induced MT depolymerization to occur, depolymerization requires the breaking of the lateral M-loop in one beta-tubulin and the H1-S2-loop in a neighboring beta-tubulin bond in side by side protofilament (PFs) [Nogalas *et al.* 1999, Nogalas *et al.* 2003, Mitra and Sept 2008]. However, there is a kinetic barrier to breaking the PF-PF bonds (i.e. with divalent cations replacing Arg 282 (+1) in the M-loop in its interactions with Glu53 (-1) in the H1-S2-loop on adjacent tubulin dimers) with the energy barrier decreasing with increasing concentration of divalent cations [Ojeda-Lopez *et al.* 2014]. This is the origin of the time-dependence for the transition between the wide-spacing and intermediate spacing MT bundle states.

Text added to manuscript (in section entitled: Time-dependent SAXS generates kinetic phase diagrams revealing three distinct assembly structures for ab-tubulin/tau/divalent cation/GTP mixtures at 37°C.).

Kinetic phase diagrams for Ca^{2+} (Fig. 3A) and Mg^{2+} (Fig. 3B) summarize the SAXS data and visualize distinct regions where the B_{ws} (green), B_{int} (yellow), and (tau-coated) tubulin rings (magenta) are dominant. This data reveals a clear decrease in the lifetime of the B_{ws} with increased divalent cation content, that is likely related to a similar effect by tetra-valent spermine on paclitaxel-stabilized MTs (42), where ion induced depolymerization, depolymerization occurs from the disruption of the lateral bond due to the divalent cation between the M-loop on one b-tubulin and the H1-S2-loop on the neighboring b-tubulin [Ojeda-Lopez 2014]. This effect was not observed with increased monovalent cations added to standard buffer, instead showing that d_{w-w} remained constant over the tested range of added KCl (up to 150mM KCl added to the PIPES buffer, Fig. 4).

3.10 The phase diagram was generated assuming that for any given ion concentration and time there are no mixtures of states (except for the intermediate helical lattice that is presented as coexisting with rings). It is not obvious to me that rings are not present together with the wide spaced lattice of MTs. Also, because the “Bragg” peaks are not obvious for 1.2 mM Ca^{++} at T_0 , it is hard to say that the shorter helical repeat was not present there already. For longer time frames at that concentration, there seem to be no obvious helical lattice at 6 or 12 hours, and only appears at a time when few MTs are still present.

With regard to the reviewer's first concern regarding rings being present in the B_{ws} state:

We agree that rings are present during the wide-spacing state, and even point to them in our TEM images in Figure 10. It is known that tubulin oligomers, rings, dimers, and MTs are found in equilibrium with one another. Our claim is that the fraction of tubulin polymerized within the MT lattice is many times higher than the fraction of tubulin existing as curved tubulin oligomers for the B_{ws} state. This is in contrast to the B_{int} state, where the scattering contribution from both tubulin rings and tubulin oligomers are appreciable. We have changed the following sentences to make this more clear:

Updated text to manuscript (in section entitled: Time-dependent SAXS generates kinetic phase diagrams revealing three distinct assembly structures for ab-tubulin/tau/divalent cation/GTP mixtures at 37°C.)

Old text:

While the MT bundles and curled tubulin oligomers are the dominate structures of the B_{ws} and ring state respectively we note that he B_{INT} state is moreso a two phase system of bundled MTs and dublin rings, where scattering contributions of both bundled MTs and tubulin rings can be resolved.

New text:

We note that while curved tubulin oligomers and MTs exist to some degree in all three labeled phases, MT bundles and curved tubulin oligomers/rings are the dominate structures of the B_{ws} and ring state, respectively, whereas in the B_{int} state scattering contributions from rings become clearly observed in addition to scattering from MT bundles.

With regard to the reviewer's second concern that the B_{int} spacing might have always existed:

We know that intermediate-spaced hexagonal bundles hadn't always existed (and only become "visible" once isolated and weakly bound MTs depolymerize) by looking at the raw scattering data without offset. The measured scattering profile for the B_{MT} state is the sum of the scattering intensities from different bundles. Thus if the hexagonal bundles had existed at t_0 and the reported "peak shift of the intermediate state" is nothing more than preferential depolymerization of non-hexagonally bound MTs, then (i) the scattering intensity at the intermediates state's q_{10} peak must be lower than the measured scattering intensity at that q in the wide spacing, and (ii) the well-defined off-axis diffraction peaks (such as the (1,1)) in the intermediate state would be visible/well-defined in the wide spacing state. However this isn't the

case. Below (left) is the scattering profile you refer to of the 1.2 mM Ca^{2+} sample plotted in figure 2C at 12 and 15 hours without offset. As you can see, the scattering intensity of the B_{MT} state at the 15-hour $q_{1,0}$ location is higher than at 12 hours—implying that the lattice parameter is actually changing for bundles during the transition.

We note that this same effect can be seen in the paper with Figure 7D (1.8 mM added Mg^{2+} plotted without offset).

3.11 Interestingly, it is halfway down the paper that we are told the initial time points experience a general “relaxation from a centrifugation step”. So much is concentrated on the change in the packing of a hexagonal lattice of MTs (a lattice that they have not been able to visualize by EM), and these kinds of extraneous effects are happening on the side?

As discussed in the paper, the relaxation due to centrifugation was only observed for samples in the wide-spacing state. Moreover, the text makes it very clear that the intermediate state is signaled by a rapid decrease in lattice parameter size over a relatively short period, and simultaneously a significant decrease in peak width (implying larger bundles) and increase in prominence of the off axis hexagonal Bragg peaks (e.g. (1,1) peak). Furthermore, the sudden decrease in spacing precisely correlates, in time, with the rapid onset of a large scattering contribution from free tubulin oligomers and rings (i.e. filling in of the first minimum). The combination of all of these factors makes it very easy to distinguish between the two states regardless of d_{w-w} .

3.12 The EM images do not show the presence of helical arrangements of MTs. Also, the amount of MTs present for the two time points appears to be the opposite of what SAXS experiments indicate (there are more, not less MTs at later times). The packing of MTs at 18 hours in

addition to not being hexagonal (order is missing), shows an average wall to wall distance between MTs that appears smaller than the one deduced from SAXS, although there is a very large deviation from the average. In general, SAXS and EM experiments appear to show poor correspondence. This may be due to lack of ultrastructure preservation during the EM sample preparation procedures. Obviously, cryo-ET would have been a better method although much more involved. Alternatively, the assumption of a hexagonal lattice as a major contributor to the SAXS profile is simply wrong. Indeed, there is but a hint of Bragg's peaks for the “wide lattice” and when the clearer intermediate lattice is obvious, MTs are already not the most abundant species.

The plastic-embedded TEM images are consistent with the observed SAXS patterns. We emphasize that XRD is much more sensitive at detecting periodicities than the human eye. Looking at the fitted result of the 1.8 mM Mg^{2+} SAXS pattern at $t_0 + 18$ hours (plotted in Figure 7C, and chosen as it is the same time and concentration of the EMs plotted in Figure 6D-F), multiple hexagonal peaks are visible yet the measured domain size of the hexagonal crystal is ~ 212 nm—which translates to only 3.76 MT interaxial distances in diameter (SI Figure 3)! Thus, for any given MT, we would expect MTs farther than 1.88 MT bond lengths away (i.e., a_h) to have no hexagonal correlation with it, and, furthermore, for numerous lattice defects below this length to exist. This is reasonably similar to the parallel plastic-embedded EM images shown in Figure 6E, where by eye, hexagonal bundles are roughly the size of the 200 nm scale bar.

Hand measuring the coordinate position for every MT visible within figures Figure 6A and 6B ($n = 7,755$), and then calculating the pair distance distribution function for them resulted in the following graph:

From the PDF we can see a clear second coordination shell for the intermediate state whereas only the first for the wide-spacing. This supports our SAXS data which show weak off-axis hexagonal peaks in the wide-spacing state and thus imply that the bundles are more likely to be branched strings than true hexagonal bundles. Additionally, we see that the 3 hour and 18 hour

waves approach 1 at ~95 nm and ~110 nm respectively, which is very close to the bundle radius measured through SAXS.

The maxima of the two resulting PDFs are 51.0 nm ($d_{w-w} = 21.0$ nm) and 58.1 nm ($d_{w-w} = 28.1$ nm) for 3 hours and 18 hours respectively. Both numbers are admittedly lower than the range of d_{w-w} 's typically observed and given within the text. However, this level of discrepancy is very commonly observed and expected due to the embedding process. Because of this, the plastic-embedded EM images are not presented within the paper to be a quantitative measurement of wall-to-wall spacings (as that is the purpose of our SAXS experiments). Rather, the EM images are presented as a real-space visualization of the types of lattice architectures observed in SAXS and the changes to those architectures upon phase transition, giving us positive feedback for the fitting models used to analyze reciprocal-space SAXS measurements.

Regarding your concern of more MTs existing in the 18 hour than the 3 hour image (which contradicts our SAXS results), a single slice containing a higher number density of MTs in the intermediate state compared to the wide-spacing state is not a reflection of the total number of microtubules within each state. We note that there is an innate bias in selecting fields of view that contain MT bundles when taking TEM images of MT bundles (as the structure of those bundles is what we are interested in). What is not shown are empty fields of view lacking microtubules. Our TEM images do generally show more MTs within a given phase-separated MT ensemble (as represented in the paper), however less of these ensembles were found, and thus the results are consistent with SAXS.

While we agree that Cryo-ET would have been a better method, for the reasons stated above and in the text, we feel that the plastic-embedded TEM images are a suitable qualitative complement to the respective quantitative SAXS data and merit publication.

3.13 This statement makes no sense: "This apparent preference for 1-dimensional MT bundling, despite close lateral proximity of the linear arrays, may be due to broken cylindrical symmetry with tau distributed non-uniformly on the MT surface, consistent with reports that tau forms phase-separated complexes on the surface of MTs." There is no relationship between a 1-dimensional array and the "phase-separated" complexes the authors are referring to. - In Fig. 7, could they show the form factor for rings in addition to the one for MTs? Maybe in panel D? The SAXS profiles could benefit from plotting $I \times Q$ or even $I \times Q \times Q$, especially in Figure 7. What are the dash blue lines? The yellow fits show significantly more features (resolved peaks) than the experimental curves. Again, displaying $I \times Q$ or $I \times Q \times Q$ in the y axis should help better see the similarities and differences.

We have removed the sentence. The blue dashed lines represent the fitted scattering contribution from non-MT polymerized tubulin plus background. The suggested Kratky plot was not added as the plot would not be very illuminating considering our samples are mixed phase containing multiple structure and form factors.

3.14 What do the authors mean by “whole-mount TEM image”? This is an unusual expression. Reading the Methods it is just the negative stain visualization of diluted samples.

Whole mount TEM is a term commonly used to indicate that the samples imaged were not fixed and sectioned but applied in a solution state directly onto a grid for imaging. For whole mount samples, microtubules are in solution within our reaction mixtures, and the reaction mixture is pipetted directly onto the grid. The buffer is then wicked off, and the grids are dried prior to imaging in vacuum. We use the phrase here to differentiate from our plastic-embedded TEM images, where only partial sections of an entire sample are imaged following sample fixation and embedding of our microtubule reaction mixtures.

3.15 This statement also lacks sense: “despite indications of gradual MT depolymerization due to ongoing partially suppressed dynamic instability”. What is suppressing dynamic instability with time? Why will this not be compatible with gradual depolymerization?

Dynamic instability is being partially suppressed due to the presence of the MT-stabilizing protein tau in the presence of abundant GTP. We agree that the sentence was worded awkwardly and has been updated to “despite indications of gradual MT depolymerization over time.”

3.16 Where did the radius of the ring of 16.3 nm come from? In the discussion the authors talk about tau-coated tubulin rings, which by necessity would be of a heterogeneous nature. When modeling the scattering, how was this considered?

Scattering from samples that had completely depolymerized into the ring state (and originated in the wide-spacing state) were fit to the theoretical scattering profile of tubulin rings. There was little variance in the fitted inner-diameter between samples, with the average measured inner-radius being 16.3 +/- 0.3 nm. This is covered in the methods section, and we have added a couple sentences into the text to make it more clear.

With regard to ring heterogeneity, scattering profiles in the ring state were originally modeled to a population of rings containing a gaussian distribution of radii. However, heterogeneity in ring diameter was found to only vary by ~5% and resulted in fits qualitatively similar to those without a distribution. This is because a gaussian heterogeneity in ring sizes only effectively smoothens the observed Bessel function oscillations; whereas the overall shape (R_g , mass/surface fractal dimensionality), amplitude (i.e., A_{ring}), and local maxima, are nearly identical to a population of homogeneous rings. Thus, in order to reduce the computation time and number of fitting parameters, fits presented were instead modeled to a homogeneous ring population.

3.17 They also state: “Enhanced cross bridging simultaneously decreases d_w while increasing the bundle domain size in the Bint state.” I would propose that they increase the order, not the size, specially when this is happening in a context of MT depolymerization

The domain size of a bundle/crystal is defined as twice the radial distance from a given scattering center to where other scatterers are no longer hexagonally correlated with it. Thus increased order within a bundle directly increases the “coherent” domain size of the bundle and your proposition is exactly what we were trying to convey in the quoted sentence.

3.18 Another statement, in this case in the discussion, that does not seem to have any firm experimental basis but be speculative is “(due to Mg^{2+} or Ca^{2+} mediated MT depolymerization of a fraction of MTs, either isolated or at the periphery of bundles where fewer cross-bridges to neighboring MTs exist)” In what data is this based? Is this a “likely” deduction?

This statement is deduced from changes in line shape of the q_{10} peak over time. While we will quantitatively discuss and model the deduction in a future paper where there is more space, we know this is happening for the following reason. The B_{MT} scattering pattern is a sum of thousands of differing MT configurations—ranging from singular MTs to large pseudo-hexagonal ensembles. However, the scattering pattern of small bundles and isolated MTs compared to large hexagonal bundles are very different, and a preferential depolymerization of smaller bundles would result in greater suppression at the tails of the q_{10} , which is what we see.

To avoid confusion, we have removed the statement “either isolated or at the periphery of bundles where fewer cross-bridges to neighboring MTs exist” in the revised text.

3.19 The authors say that the present results clarify “contradictory results” concerning the fact that “recent studies show that the MT-stabilizing drug paclitaxel, which severely reduces free tubulin, oligomers (at paclitaxel/tubulin-dimer molar ratios of $Apaclitaxel=1/1$), suppresses MT bundling by all six tau isoforms⁵”. There are actually TWO papers (2009 & 2017, so hardly “recent”) both of which are from the authors themselves (!), that hardly justify the claim of contradictory literature the authors refer to. Interestingly, the authors explained the effect of taxol in disrupting tau-induced MT bundling in those papers with models distinct from the one proposed here.

We have added multiple citations from outside our group to make the contradictory results statement hold merit. The statement given to why tau could stabilize MT bundles at low concentrations of PTX within the two papers cited was presented as a possible explanation within the lens of the currently accepted mechanism for MT bundling. However, with the added knowledge of our current research, bundling through multiple interpenetrating PDs seems less valid; and as such, we attempt to explain previous results through our current model.

Text added/clarified in manuscript:

The model reconciles years of contradicting reports regarding tau's role in bundling. While numerous early publications pointed to MT bundling as one of many roles of tau^{7,8,11,53}, several cell free studies of MTs containing the MT-stabilizing drug paclitaxel pointed to tau's apparent inability to mediate MT bundles [new refs]. More recent SAXS studies show that paclitaxel, at paclitaxel/tubulin-dimer molar ratios of $\Lambda_{\text{paclitaxel}}=1/1$, suppresses MT bundling by all six tau isoforms⁵⁶. Follow-up SAXS and TEM experiments showed that reducing paclitaxel below $\Lambda_{\text{paclitaxel}} \approx 1/8$, restores tau-mediated MT bundles⁵⁷. These findings are consistent with our central discovery that bundling of MTs by tau requires free tubulin oligomers (i.e. where $\Lambda_{\text{paclitaxel}}=1/1$ [56] severely reduced free tubulin oligomers and reducing paclitaxel below $\Lambda_{\text{paclitaxel}} \approx 1/8$ [57] restored free tubulin).

1. Choi, M. C., et al. "Human microtubule-associated-protein tau regulates the number of protofilaments in microtubules: a synchrotron x-ray scattering study." *Biophysical Journal* 97.2 (2009): 519-527.
2. Duan, Aranda R., et al. "Interactions between tau and different conformations of tubulin: implications for tau function and mechanism." *Journal of Molecular Biology* 429.9 (2017): 1424-1438.
3. Li, Xiao-Han, Jacob A. Culver, and Elizabeth Rhoades. "Tau binds to multiple tubulin dimers with helical structure." *Journal of the American Chemical Society* 137.29 (2015): 9218-9221.
4. Li, Xiao-Han, and Elizabeth Rhoades. "Heterogeneous tau-tubulin complexes accelerate microtubule polymerization." *Biophysical journal* 112.12 (2017): 2567-2574.
5. Kanaan, Nicholas M., et al. "Liquid-liquid phase separation induces pathogenic tau conformations in vitro." *Nature communications* 11.1 (2020): 2809.
6. Ojeda-Lopez, Miguel A., et al. "Transformation of taxol-stabilized microtubules into inverted tubulin tubules triggered by a tubulin conformation switch." *Nature materials* 13.2 (2014): 195-203.
7. Zempel, Hans, et al. "Axodendritic sorting and pathological missorting of Tau are isoform-specific and determined by axon initial segment architecture." *Journal of Biological Chemistry* 292.29 (2017): 12192-12207.
8. Mitra, Arpita, and David Sept. "Taxol allosterically alters the dynamics of the tubulin dimer and increases the flexibility of microtubules." *Biophysical journal* 95.7 (2008): 3252-3258.

9. Nogales, Eva, et al. "High-resolution model of the microtubule." *Cell* 96.1 (1999): 79-88.
10. Nogales, Eva, Hong-Wei Wang, and Hanspeter Niederstrasser. "Tubulin rings: which way do they curve?." *Current opinion in structural biology* 13.2 (2003): 256-261.

REVIEWERS' COMMENTS

Reviewer #2 (Remarks to the Author):

The authors have addressed most of my concerns raised in the review. However, two particular points still need attention:

1) Regarding point 2.5, articles reporting the multivalent salt effects include <https://journals.aps.org/prl/abstract/10.1103/PhysRevLett.91.028301> or <https://onlinelibrary.wiley.com/doi/pdf/10.1002/prot.22852>.

2) Concerning point 2.3, while the SAXS data fittings demonstrate an impressive quality and the analysis/interpretation is satisfactory, the absence of tabulated fitting parameters is a notable gap. It would significantly enhance the clarity and depth of understanding of each scattering component—including their relative contributions to the data and their sizes—if these fitting parameters were presented, at least in the supporting information.

Reviewer #3 (Remarks to the Author):

I commend the authors for the added explanations, added experiments and clarifications. In my opinion, now that I have also seen the comments of the other reviewers, the paper appears dense, highly specialized in its methodology and analysis, and with somehow weak connections to what may be happening in vivo. On the other hand, the amount of work is truly remarkable and the value of the proposed model to be seen and assessed by the microtubule community and tested by alternative approaches is not insignificant. While I cannot be truly enthusiastic of the work, I am not opposed to its publication.

Reviewer: 2 (in *italics*)

1) Regarding point 2.5, articles reporting the multivalent salt effects include <https://journals.aps.org/prl/abstract/10.1103/PhysRevLett.91.028301> or <https://onlinelibrary.wiley.com/doi/pdf/10.1002/prot.22852>.

Our Response:

The two papers cited by the reviewer describe “disorder-order-disorder” re-entrant phase behavior due to charge inversion of macro-ions (charged proteins and filamentous proteins) as a function of increasing concentrations of multivalent counter-ions. The systems described in the paper are (i) at “thermal equilibrium”, and (ii) in the strong coupling regime of electrostatics where counter-ion condensation on macro-ions (i.e. Manning condensation) leads to short-range attraction and self-assembly of macro-ions (with the range of the attraction close to the size of the counterion, i.e., sub-nanometers). Further increases in the multivalent ion concentration leads to charge-inversion of the macro-ion, repulsions between macro-ions, and thus disassembly.

The kinetic phase diagrams reported in our paper involve an “out-of-equilibrium” transition between two “ordered phases” of bundled microtubules (MTs) (i.e. the wide-spaced and intermediate spaced bundled phases) undergoing cycles of growth and depolymerization, where MT depolymerization is eventually observed either above a certain concentration of divalent ions or as a function of time at lower concentrations (consistent with the key hypothesis that divalent ions favor the depolymerization state of MT dynamic instability). There is no re-entrant behavior and no charge-inversion of our macro-ions (i.e. the microtubules).

2) Concerning point 2.3, while the SAXS data fittings demonstrate an impressive quality and the analysis/interpretation is satisfactory, the absence of tabulated fitting parameters is a notable gap. It would significantly enhance the clarity and depth of understanding of each scattering component—including their relative contributions to the data and their sizes—if these fitting parameters were presented, at least in the supporting information.

Our Response:

We agree with the reviewer. We have added all the fitting parameters related to all of the SAXS data in the source data file.

Reviewer: 3

The reviewer had no further comments on the revised manuscript